# Dense Connector for MLLMs

**Huanjin Yao**[1,3*], **Wenhao Wu**[2* ✉], **Taojiannan Yang**[4], **Yuxin Song**[3], **Mengxi Zhang**[3]
**Haocheng Feng**[3], **Yifan Sun**[3], **Zhiheng Li**[1 ✉], **Wanli Ouyang**[5], **Jingdong Wang**[3]
[1]Shenzhen International Graduate School, Tsinghua University    [2]The University of Sydney
[3]Baidu Inc.    [4]Amazon    [5] The Chinese University of Hong Kong
* Equal Contribution    ✉ Corresponding Author

## Abstract

*Do we fully leverage the potential of visual encoder in Multimodal Large Language Models (MLLMs)?* The recent outstanding performance of MLLMs in multimodal understanding has garnered broad attention from both academia and industry. In the current MLLM rat race, the focus seems to be predominantly on the linguistic side. We witness the rise of larger and higher-quality instruction datasets, as well as the involvement of larger-sized LLMs. Yet, scant attention has been directed towards the visual signals utilized by MLLMs, often assumed to be the final high-level features extracted by a frozen visual encoder. In this paper, we introduce the ***Dense Connector*** - a simple, effective, and plug-and-play vision-language connector that significantly enhances existing MLLMs by leveraging multi-layer visual features, with minimal additional computational overhead. Building on this, we also propose the Efficient Dense Connector, which achieves performance comparable to LLaVA-v1.5 with only 25% of the visual tokens. Furthermore, our model, trained solely on images, showcases remarkable zero-shot capabilities in video understanding as well. Experimental results across various vision encoders, image resolutions, training dataset scales, varying sizes of LLMs (2.7B→70B), and diverse architectures of MLLMs (*e.g.*, LLaVA-v1.5, LLaVA-NeXT and Mini-Gemini) validate the versatility and scalability of our approach, achieving state-of-the-art performance across 19 image and video benchmarks. We hope that this work will provide valuable experience and serve as a basic module for future MLLM development. Code is available at `https://github.com/HJYao00/DenseConnector`.

## 1 Introduction

In recent years, Large Language Models (LLMs) led by ChatGPT [1] have made remarkable advancements in text comprehension and generation. Furthermore, cutting-edge Multimodal Large Language Models (MLLMs) [2, 3] have rapidly expanded the capabilities of LLMs to include visual understanding, evolving into models capable of integrating both vision and text modalities. This has elevated MLLMs to become a new focal point for research and discussion [4, 5, 6, 7].

In broad terms, the architecture of existing MLLMs can be delineated into three components: the pre-trained vision encoder (*e.g.*, CLIP's ViT-L [8] or EVA-CLIP's ViT-G [9]), the pre-trained LLM (*e.g.*, OPT [10], Llama [11], Vicuna [12], *etc.*), and the connector (*e.g.*, Q-former [13, 14] or linear projection [15, 16]) trained from scratch to bridge the vision and language models. An intriguing trend in current MLLM research is that the focus of model learning and performance improvement seems to primarily center around the language aspect (*e.g.*, utilizing larger-scale and higher-quality visual instruction data [17, 16, 18], larger-sized LLMs [19, 20]), with less exploration into the visual signals fed into the connector. Typically, the visual encoder is frozen to extract high-level visual features, which are then fed into the connector. This leads us to rethink: *Have we fully utilized the existing pre-trained visual encoder?*

38th Conference on Neural Information Processing Systems (NeurIPS 2024).

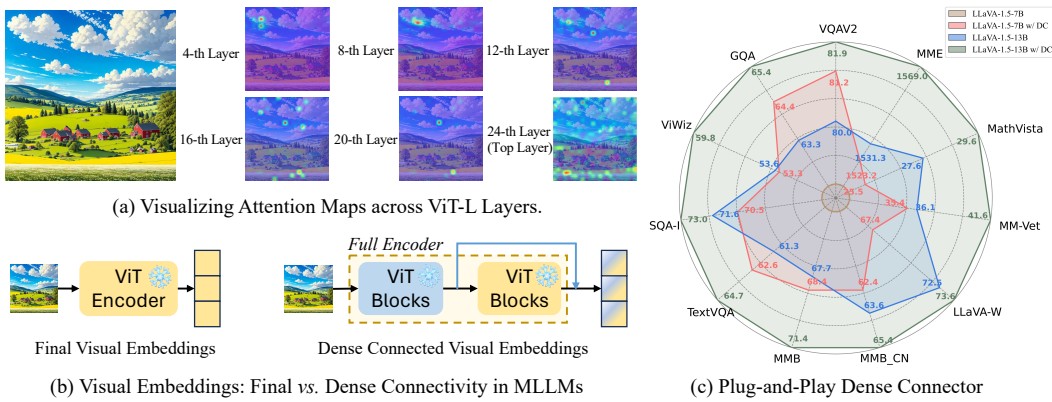

Figure 1: Exploring Multi-layer Visual Features Empowering existing MLLMs.

In addition to the common practice of feeding the connector with final high-level visual features from visual encoder, an intuitive yet overlooked idea is to integrate visual features from various layers to complement the high-level features. In Fig. 1 (a), we illustrate attention maps from different layers of a 24-layer CLIP [8] pre-trained ViT-L [21], showing that different layers of the same visual encoder emphasize different regions of interest. Moreover, looking back at the history of computer vision, classic models (*e.g.*, Densenet [22], FPN [23]) utilize multi-layer features to enhance visual representations. In MLLMs, the vision encoder is typically frozen to mitigate significant computational costs. In this context, our idea leverages the "free lunch" of utilizing offline features from different layers as an implicit enhancement of visual information without the need for additional computational overhead. Furthermore, this way also complements techniques that directly increase visual signals, *e.g.*, increasing image resolution [18, 24, 25, 26, 27, 28] or introducing additional visual encoders [29, 30, 18]. This idea is both simple and efficient, while also being sufficiently generic, logically allowing for seamless integration with any existing MLLMs.

In light of this, we propose the ***Dense Connector (DC)***, serving as a plug-and-play vision-language connector that involves offline features from various layers of the frozen visual encoder to provide the LLM with more visual cues. We explore three intuitive instantiations for the Dense Connector: 1) *Sparse Token Integration (STI)*: We explicitly consider increasing the number of visual tokens by aggregating visual tokens from different specified layers and the final visual token. These tokens are then fed into a learnable projector for mapping into the text space. 2) *Sparse Channel Integration (SCI)*: To avoid increasing the number of tokens, we concatenate visual tokens from different specified layers in the feature dimension. They are then passed to the projector, which not only maps visual tokens into the text space but also serves to reduce the feature dimensionality. 3) *Dense Channel Integration (DCI)*: In addition to incorporating features from specified layers, we further attempt to utilize visual features from all layers. All of these instantiations yield significant improvements while utilizing just one simple learnable projector (comprising two linear layers) without introducing any extra parameters. Moreover, we conduct extensive empirical studies to demonstrate its scalability and compatibility. We summarize our contributions as follows:

- We propose a simple, effective, and plug-and-play *Dense Connector* that enhances the visual representation of existing MLLMs with minimal additional computational overhead. We hope it can serve as a basic module to continuously benefit future MLLMs.

- We demonstrate the versatility and scalability of our approach across various visual encoders, image resolutions (336px→768px), training dataset scales, varying sizes of LLMs (2B→70B), and diverse MLLMs architectures (*e.g.*, LLaVA-v1.5 [16], LLaVA-NeXT [25], Mini-Gemini [18]).

- Our method exhibits exceptional performance across 11 image benchmarks and achieves state-of-the-art results on 8 video benchmarks without the need for specific video tuning.

## 2 Related Work

### 2.1 Large Pre-trained Vision Models

The advent of pre-trained Vision Transformers (ViT) [21] has significantly propelled the advancement of computer vision. Furthermore, pre-training ViT models on web-scale image-text pairs, *e.g.*, CLIP [8] and its subsequent iterations [9, 31, 32, 33], where vision and text encoders are simultaneously trained to bridge the gap between visual and textual modalities, has introduced zero-shot visual perception capabilities. Since then, CLIP-like models have served as effective initializations and have been incorporated into various vision-language cross-modal models (*e.g.*, video-text alignment [34, 35, 36, 37], large vision-language models [14, 15, 38], *etc.*). Recently, SigLIP [31] introduced pairwise sigmoid loss during training, enabling the visual encoder to demonstrate more advanced visual perception capabilities. To validate the compatibility of our *Dense Connector*, this paper conducted experiments on different visual encoders, including those of CLIP [8] and SigLIP [31].

### 2.2 Large Language Models

The exceptional text understanding and generation capabilities demonstrated by auto-regressive Large Language Models (LLMs) [39, 40, 41] have garnered significant attention. Subsequently, a plethora of LLMs [42, 11, 10, 43] have emerged, with notable open-source efforts like LLaMA [42] greatly propelling community contributions to LLMs research. Through instruction fine-tuning techniques, these models showcase human-like language interaction abilities, further further propelling advancements in natural language processing. Recent developments have seen LLMs scaled up or down to meet various application needs. Lightweight LLMs [44, 45, 20, 46] have been developed to address computational constraints, facilitating edge deployment. Conversely, in the pursuit of exploring the upper limits of LLMs, works such as [47, 19, 11, 20] have expanded LLM parameters, continuously pushing the boundaries of language capabilities. In this study, we validated the scalability of our *Dense Connector* by employing multiple LLMs ranging from 2.7B to 70B parameters.

### 2.3 Multimodal Large Language Models

After witnessing the success of LLMs, researchers have shifted their focus towards enabling LLMs to understand visual signals. To achieve this, prior research has proposed compressing visual embeddings using Q-former [14] into query embeddings, followed by transforming them into text embeddings through linear projection, or directly employing MLP projection [15] to connect the visual encoder with LLM. Furthermore, following the instruction tuning paradigm [48, 49], pioneering works [38, 15, 50] significantly boost the development of MLLMs through visual instruction tuning. Subsequently, by introducing larger-scale and higher-quality datasets, efforts such as [18, 25, 17, 24] have notably enhanced the visual understanding and reasoning capabilities of MLLMs. Additionally, there are works that introduce additional visual encoders [29, 30] or utilize higher-resolution images [25, 18, 24] to provide richer visual signal sources. Meanwhile, many studies [51, 52, 53] directly extend these above image-based methods to video conversational models by leveraging video instruction tuning datasets. In summary, these studies typically utilize high-level features from the frozen visual encoder as visual embeddings. However, we find that effectively leveraging offline features from different layers—the overlooked "free lunch"—can yield significant benefits. Then, we follow FreeVA [54] to directly extend the image model for video understanding without any additional video training.

## 3 Method

### 3.1 Overview

In Fig. 2(a), we illustrate the overall architecture of our model, using the mainstream LLaVA [15] framework as an example. It includes the pre-trained visual encoder $Vis(\cdot)$ and the Large Language Model $LLM(\cdot)$, alongside with our proposed *Dense Connector* $DC(\cdot)$. Similarly, our *DC* can be seamlessly extended to other high-resolution or dual-branch MLLMs, such as LLaVA-NeXT [25] and Mini-Gemini [18]. Formally, the introduction is as follows:

**Visual Encoder:** We utilize a CLIP pre-trained Vision Transformer (ViT) [21] as the visual encoder for extracting visual features. Initially, ViT partitions an image $X_i \in \mathbb{R}^{H \times W \times C}$ into a sequence of

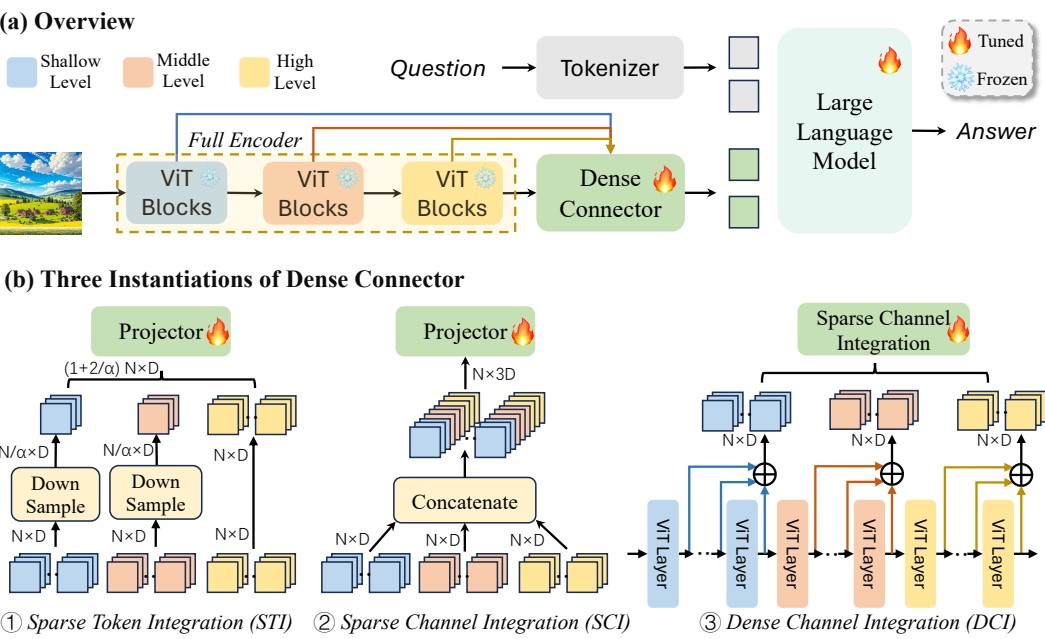

Figure 2: Dense Connector in MLLM: Overview and Three Instantiations. $N$ is the number of tokens, $D$ is the feature dimension, and $\alpha$ is the downsampling ratio.

non-overlapping patches. Each patch is then processed via convolution to produce visual tokens, which are subsequently input into ViT. This procedure yields $L$ layers of visual features $V \in \mathbb{R}^{L \times N \times D_v}$, where $N$ denotes the number of the visual tokens and $D_v$ denotes the feature dimension.

**Dense Connector:** The Dense Connector comprises two components: the first integrates multi-layer visual features, elaborated upon in detail in Sec. 3.2, while the second employs a learnable MLP to map the integrated visual features to the LLM's text space. The MLP consists of two linear layers with a GELU [55] activation function sandwiched between them. The first layer adjusts the visual hidden size $D_v$ to align with the LLM's hidden dimension $D_t$, while the second layer maintains the dimensionality at $D_t$. Upon processing through the Dense Connector, we acquire visual embeddings $e_v \in \mathbb{R}^{N \times D_t}$ that encapsulate information from multiple layers.

**Large Language Model:** The LLM processes textual data using a tokenizer and text embedding module to convert language into its input feature space. These text embeddings are concatenated with transformed visual embeddings before being fed into the LLM for subsequent predictions.

## 3.2 Dense Connector

Here we delve into three intuitive instantiations of the **Dense Connector**, each demonstrating superior performance compared to the baseline (*e.g.*, LLaVA-1.5 [16]). Among them, *Sparse Token Integration (STI)* and *Sparse Channel Integration (SCI)* sparsely select visual features from $K$ layers (indexed as $l_n$, where $1 \leq l_n < L$ and $1 \leq n \leq K$) spanning shallow, middle, and high levels out of the total $L$ layers of ViT, while *Dense Channel Integration (DCI)* utilizes features from all layers. These features are then fed into Dense Connector to generate visual embedding that can be "understood" by LLM.

**Sparse Token Integration (STI):** While existing methods typically rely solely on features from the final layer as the visual representation input for the LLM, our STI approach diverges from this by integrating features from multiple layers to enrich the visual input for the LLM. Recognizing that higher-level features contain richer semantic information crucial for visual signal perception in VLMs, we maintain the final layer features unchanged while downsampling additional visual features from other layers by using average pooling $avg(\cdot)$ with a stride $\alpha$. This downsampling reduces the number of visual tokens to $N' = N/\alpha$, mitigating computational overhead and redundancy. These visual features from various layers are concatenated along the token dimension and processed through a shared $MLP(\cdot)$, yielding more robust visual embedding $e_v \in \mathbb{R}^{(N+(k-1)\times N')\times D_t}$:

$$e_v = MLP(Concatenate([avg(V_{l_1}), ..., avg(V_{l_K}), V_L], dim = token)). \tag{1}$$

**Sparse Channel Integration (SCI):** We delve deeper into connecting multi-level features along the channel dimension. Subsequently, this feature is processed through an MLP projector to obtain visual embedding $e_v \in \mathbb{R}^{N \times D_t}$:

$$e_v = MLP(Concatenate([V_{l_1}, ..., V_{l_K}, V_L], dim = channel)). \tag{2}$$

The MLP projector serves dual functions: integrating various features as a fusion tool and facilitating the transformation of visual inputs into linguistic representations. This design ingeniously leverages the dimensionality scaling effect of the MLP projector, enabling the transformation of connected multi-layer features into the feature space of the LLM without requiring additional modules. Moreover, this method does not increase the number of tokens fed into the LLM, thereby avoiding any increase in the computational overhead of the LLM.

**Dense Channel Integration (DCI):** While *Sparse Channel Integration* incorporates features from $K$ layers, many visual feature from other layers remain unused. Concatenating all visual feature layers using STI or SCI leads to excessively high dimensions, posing challenges during training. To address these issues, we propose DCI, which builds upon the SCI method by integrating adjacent layers to reduce redundancy and high dimensionality. This approach ensures dense connectivity across a wider range of visual layers. Specifically, we partition the features of $L$ layers into $G$ groups, where each group comprises $M$ adjacent visual features, with $M = L/G$. Summing the features within each group, denoted as $GV_g$, finally yields $G$ fused visual representations:

$$GV_g = \frac{1}{M} \sum_{i=(g-1)M+1}^{gM} V_i, \quad 1 \le g \le G. \tag{3}$$

Subsequently, we concatenate these features from the $G$ groups with the final layer's features along the channel dimension before passing them through an MLP:

$$e_v = MLP(Concatenate([GV_1, ..., GV_G, V_L], dim = channel)). \tag{4}$$

### 3.3 Efficient Dense Connector for Visual Token Optimization

For MLLMs, each image is converted into hundreds or even thousands of visual tokens, and this large number of tokens increases the computational burden on autoregressive models. However, reducing the number of visual tokens generally leads to a noticeable drop in performance. In this paper, we leverage multi-layer visual features to compensate for the information loss caused by reducing visual tokens, enabling our method to achieve performance on par with LLaVA-v1.5 [16] while using several times fewer visual tokens, and surpassing other carefully designed efficient connectors [24, 38, 50, 56, 57] Specifically, after obtaining the visual embedding $e_v$ through the Dense Connector, we apply a parameter-free module, *i.e.* a 2D interpolation function, to downsample visual tokens. Then, these discrete visual tokens $e'_v$ are concatenated with the text tokens and fed into the LLM, resulting in a 3 times improvement in inference speed.

### 3.4 Training-Free Extension from Image to Video Conversational Models

Following FreeVA [54], we extend the image-based models trained as described above to video domain for video understanding. Specifically, given a video, we uniformly sample $T$ frames, and each frame is processed through the visual encoder and dense connector to obtain a visual embedding $e_v$. Consequently, we obtain an embedding sequence $\{e_{v_1}, ..., e_{v_T}\}$, which is then fed into the LLM.

# 4 Experiments

## 4.1 Implementation Details

**Architecture.** 1) Visual Encoders: To explore the generalization of the Dense Connector across different visual encoders, we select two mainstream options, namely CLIP-ViT-L-336px [8] and SigLIP-ViT-SO [31]. 2) LLMs: The Dense Connector is applied across various LLMs, spanning from 2.7B to 70B parameters. This includes Phi-2-2.7B [44], Vicuna-7B&13B [12], Hermes-2-Yi-34B [19], and Llama3-8B&70B-Instruct. 3) Dense Connector: For the 24-layer CLIP-ViT-L-336px [8], we specifically target visual features from the 8th, 16th, and final 24th layers for both STI and SCI. For STI, we apply a downsampling factor of $\alpha = 8$ for the features from the 8th and 16th layers. For DCI, we divide all layer features into two groups, each containing 12 layers (*i.e.*, 1-12, 13-24). Note that we use 1 to denote the stem layer of ViT, *e.g.*, 2-25 correspond to the 24 layers of ViT-L. Similarly, for the SigLIP ViT-SO [31], which has 27 layers, we partition the first 26 layers into two groups (*i.e.*, 1-13, 14-26).

**Training Datasets.** Data quality plays a crucial role in determining the performance of MLLMs. In this study, we examine the impact of two high-quality training datasets on our model: LLaVA-1.5 [16] and Mini-Gemini [18]. The LLaVA-1.5 pre-training dataset comprises 558K image captions, while its instruction tuning dataset contains 665K conversations. Mini-Gemini builds upon LLaVA-1.5, offering a larger dataset with 1.2M image-text caption pairs for alignment and 1.5M conversations for instruction tuning. Unless otherwise specified, all experimental results are based on the LLaVA-1.5 dataset to reduce training costs.

**Training Recipe.** We train all models on 8 NVDIA A100 GPUs with 40GB VRAM, except for the Hermes-2-Yi-34B and LLama-3-70B-Instruct, which utilize 32 NVDIA A100 GPUs with 80GB VRAM. Our training process comprises two stages: pre-training and instruction fine-tuning. In the pre-training phase, we initialize the visual encoder and LLM with pre-trained weights, while the Dense Connector is randomly initialized. Here, we freeze the visual encoder and the LLM, updating only the parameters of the Dense Connector. The model undergoes pre-training for one epoch with a global batch size of 256 and a learning rate of 1e-3. Subsequently, in the instruction fine-tuning stage, we maintain the visual encoder frozen while updating the Dense Connector and the LLM. Fine-tuning is performed for 1 epoch with a global batch size of 128 and a learning rate of 2e-5. For models using LoRA fine-tuning, we set the LoRA rank to 128 and LoRA alpha to 256. When scaling up the LLM to larger parameter sizes, such as LLama-3-70B-Instruct, we apply LoRA fine-tuning due to memory constraints. In this setup, we set the LoRA rank to 128 and LoRA alpha to 256.

**Evaluation.** We present comprehensive results across various image and video evaluation benchmarks. For image datasets, we include GQA [58], VQAV2 ($VQA^{v2}$) [59], ScienceQA ($SQA^I$) [60], TextVQA ($VQA^T$) [61], POPE [62], MathVista (Math) [63], MMBench (MMB) [64], MM-Vet (MMV) [65], MMMU [66], LLaVA-Bench-In-the-Wild (LBW) [15], and MME [67]. Additionally, we evaluate zero-shot performance on open-ended video question-answering benchmarks such as MSVD-QA [68], ActivityNet-QA [69], MSRVTT-QA [70], and the newly proposed generative performance benchmark [51], include evaluation metrics such as Correctness of Information (CI), Detail Orientation (DO), Contextual Understanding (CU), Temporal Understanding (TU), and Consistency (CO).

## 4.2 Ablation Study

**Study on Instantiations of Dense Connector.** In Tab. 1, we discuss three proposed methods of Dense Connector. 1) **Sparse Token Integration (STI):** This method uses visual features from different hierarchical levels as independent visual prompts for the LLM, allowing it to perceive a more diverse set of visual features. We select features from the 8th, 16th, and 24th layers, resulting in significant improvements across various datasets, particularly achieving a 2.9% increase on the MMB [64]. Further expanding the selection to include the 8th, 16th, 20th, and 24th layers enhances performance but also increases token count, leading to higher training costs and inference time. 2) **Sparse Channel Integration (SCI):** This method efficiently uses the MLP projector for feature fusion and projection, integrating multiple layers of visual features into visual embeddings. SCI boosts performance with minimal additional computational cost, achieving peak performance with features from the 8th, 16th, and 24th layers, resulting in a 1.7% improvement on GQA. SCI performs better than STI and mitigates the computational costs associated with higher token counts. However, expanding the range of visual

Table 1: Ablations on Visual Layer Selection in Dense Connector. Here, we explore three instantiations (*STI*, *SCI*, and *DCI*) of our Dense Connector integrated with the baseline (*i.e.*, LLaVA-1.5 [16]), which utilizes a 24-layer CLIP-ViT-L-336px.

| Model | Layer Index | GQA | VQA$^{v2}$ | SQA$^I$ | VQA$^T$ | POPE | MMB | MMV | LBW |
|---|---|---|---|---|---|---|---|---|---|
| Baseline | 24 | 62.0 | 78.5 | 66.8 | 58.2 | 85.9 | 64.3 | 31.1 | 65.4 |
| + *STI* | 8,16,24 | 63.3 | 79.1 | 68.0 | 58.0 | 85.8 | 67.2 | 30.9 | 65.5 |
| + *STI* | 8,16,20,24 | 63.0 | 79.1 | 68.0 | 58.8 | 85.9 | 67.6 | 30.8 | 65.7 |
| + *SCI* | 8,16,24 | 63.7 | 79.2 | 68.9 | 58.2 | 86.1 | 66.2 | 32.2 | 66.0 |
| + *SCI* | 16,24 | 63.0 | 79.0 | 67.6 | 58.2 | 86.0 | 65.6 | 31.7 | 65.6 |
| + *SCI* | 8,16,20,24 | 63.6 | 79.2 | 67.0 | 58.1 | 86.0 | 65.8 | 31.9 | 66.0 |
| + *DCI* | (1-8),(9-16),(17-24) | 63.6 | 79.3 | 67.8 | 58.6 | 86.3 | 66.5 | 32.6 | 66.0 |
| + *DCI* | (1-12),(13-24) | **63.8**$^{1.8\uparrow}$ | **79.5**$^{1.0\uparrow}$ | **69.5**$^{2.7\uparrow}$ | **59.2**$^{1.0\uparrow}$ | **86.6**$^{0.7\uparrow}$ | **66.8**$^{2.5\uparrow}$ | **32.7**$^{1.6\uparrow}$ | **66.1**$^{0.7\uparrow}$ |

Table 2: Exploring the Compatibility and Scalability of Dense Connector (DC). Scaling results on visual encoder (VE), resolution (Res.), pre-training (PT) / instruction tuning (IT) data, and LLM are provided. "0.5M+0.6M" denotes the training data from LLaVA-1.5 [16], while "1.2M+1.5M" denotes the data from Mini-Gemini [18]. $^*$ indicates results evaluated using official model.

| Method | VE | Res. | PT+IT | LLM | GQA | SQA$^I$ | VQA$^T$ | MMB | MMV | MMMU$^v$ | Math |
|---|---|---|---|---|---|---|---|---|---|---|---|
| *Scaling to more powerful visual encoder* | | | | | | | | | | | |
| LLaVA [16] | CLIP-L | 336 | 0.5M+0.6M | Vicuna-7B | 62.0 | 66.8 | 58.2 | 64.3 | 31.1 | 35.3$^*$ | 24.9$^*$ |
| LLaVA [16] | CLIP-L | 336 | 0.5M+0.6M | Vicuna-13B | 63.3 | 71.6 | 61.3 | 67.6 | 36.1 | 36.4 | 27.6 |
| DC (*w/ LLaVA*) | CLIP-L | 336 | 0.5M+0.6M | Vicuna-7B | 63.8 | 69.5 | 59.2 | 66.8 | 32.7 | 34.8 | 26.9 |
| DC (*w/ LLaVA*) | SigLIP-SO | 384 | 0.5M+0.6M | Vicuna-7B | 64.2 | 70.5 | 62.6 | 68.4 | 35.4 | **36.7** | 25.5 |
| DC (*w/ LLaVA*) | SigLIP-SO | 384 | 0.5M+0.6M | Vicuna-13B | **65.4** | **73.0** | **64.7** | **71.4** | **41.6** | 34.3 | **29.6** |
| *Scaling to larger-scale training data* | | | | | | | | | | | |
| DC (*w/ LLaVA*) | SigLIP-SO | 384 | 1.2M+1.5M | Vicuna-7B | 63.8 | 72.9 | 64.6 | 71.7 | 45.0 | 35.8 | 33.1 |
| DC (*w/ LLaVA*) | SigLIP-SO | 384 | 1.2M+1.5M | Vicuna-13B | **64.6** | **77.1** | **65.0** | **74.4** | **47.7** | **37.2** | **36.5** |
| *Scaling to high resolution with a dual visual encoder* | | | | | | | | | | | |
| MGM [18] | CLIP-L +ConvX-L | 336 +768 | 1.2M+1.5M | Vicuna-7B | 62.6$^*$ | 70.4$^*$ | 65.2 | 69.3 | 40.8 | 36.1 | 31.4 |
| MGM [18] | CLIP-L +ConvX-L | 336 +768 | 1.2M+1.5M | Vicuna-13B | 63.4$^*$ | 72.6$^*$ | 65.9 | 68.5 | 46.0 | 38.1 | 37.0 |
| DC (*w/ MGM*) | CLIP-L +ConvX-L | 336 +768 | 1.2M+1.5M | Vicuna-7B | 63.3 | 70.7 | 66.0 | 70.7 | 42.2 | 36.8 | 32.5 |
| DC (*w/ MGM*) | CLIP-L +ConvX-L | 336 +768 | 1.2M+1.5M | Vicuna-13B | **64.2** | **74.9** | **66.7** | 70.7 | **49.8** | **39.3** | **38.1** |
| *Scaling to dynamic high resolution* | | | | | | | | | | | |
| LLaVA-NeXT [16] | CLIP-L | AnyRes | 0.5M+0.6M | Vicuna-7B | 64.0 | 69.5 | 64.5 | 66.5 | 33.1 | 35.4 | 25.7 |
| DC (*w/ LLaVA*) | CLIP-L | AnyRes | 0.5M+0.6M | Vicuna-7B | 64.6 | **70.5** | 65.6 | **67.4** | 33.7 | **37.6** | 26.2 |
| DC (*w/ LLaVA*) | SigLIP-SO | AnyRes | 0.5M+0.6M | Vicuna-7B | **64.8** | 69.3 | **66.5** | 67.2 | **34.8** | 36.3 | **27.0** |

layers within SCI does not yield additional performance gains, suggesting that merely extending the range of visual feature layers is ineffective. 3) **Dense Channel Integration (DCI):** Building on the performance enhancements of SCI, DCI integrates a broader array of visual features using grouped additive fusion to produce robust visual representations. We divide the visual features into 2 or 3 groups, each with an equal number of layers. For the CLIP-L model, each group combines features from 12 or 8 layers, respectively. Splitting them into 2 groups demonstrates superior performance, achieving improvements of 2.7% on SQA [60] and 2.5% on MMB [64] compared to the baseline. The experimental results illustrate that utilizing multi-layer visual features enhances the visual perception capabilities of the MLLMs, leading to more accurate responses. Unless otherwise specified, we employ DCI as the default instantiation of the Dense Connector for optimal performance.

**Study on Visual Encoders and Training Dataset Impacts.** Given our method's reliance on multi-layer visual features, it is crucial to assess its impact across various visual backbones. As shown in Tab. 2, we first replace the CLIP-ViT-L [8] with the more advanced visual encoder SigLIP-ViT-

Table 3: Comparison of Efficient Dense Connector with Other Efficient Methods. * indicates results evaluated using official model.

| Method | Res. | #Token | PT+IT | LLM | GQA | VQA$^{v2}$ | SQA$^I$ | VQA$^T$ | MMB | MMV | Math |
|---|---|---|---|---|---|---|---|---|---|---|---|
| LLaVA [16] | 336 | 576 | 0.5M+0.6M | Vicuna-7B | 62.0 | 78.5 | 66.8 | **58.2** | 64.3 | 31.1 | 24.9* |
| Qwen-VL-Chat [24] | 448 | 256 | 1.4B+50M | Qwen-7B | 57.5 | 68.2 | 61.5 | - | - | - | - |
| TokenPacker [56] | 336 | 144 | 0.5M+0.6M | Vicuna-7B | 61.9 | 77.9 | - | - | 65.1 | 33.0 | - |
| Dense Connector | 336 | 144 | 0.5M+0.6M | Vicuna-7B | **62.8** | **79.4** | **68.8** | 58.1 | **67.6** | **34.4** | **25.8** |

SO [31]. Leveraging the enhanced multi-layer visual features from SigLIP-ViT-SO, our Dense Connector demonstrates further performance improvements. Additionally, we investigate the influence of training datasets on the effectiveness of the Dense Connector. By fine-tuning our model using the larger dataset [18], we observe notable performance gains across most benchmark evaluations. The results indicate that more training data significantly enhance model performance. Specifically, our model with Vicuna-13B achieves accuracy of 36.5% on MathVista [63] and 77.1% on SQA [60], underscoring the significant benefits of increased training data. Moreover, when using the same training data and the LLM (*i.e.*, Vicuna-7B), our Dense Connector surpasses the dual encoder structure of Mini-Gemini [18] across the majority of benchmarks. Specifically, it achieves performance gains of 2.4% on the MMB [64], 4.2% on MM-Vet [65], and 1.7% on the MathVista [63] benchmark.

**Study on High-Resolution Setting.** The use of high-resolution images to enhance detail representation in MLLMs has garnered considerable attention [25, 18, 26, 27, 28]. In this paper, we extend Dense Connector to the Mini-Gemini (MGM) [18] and LLaVA-NeXT [25], showcasing its plug-and-play capability. For MGM, we keep the high-resolution features from ConvNeXT [71] intact, applying DCI exclusively to the CLIP features. We observe significant improvements across various benchmarks, as detailed in Tab. 2, including MathVista [63], MMB [64], and MM-Vet [65], with enhancements of 1.1%, 2.2%, and 3.8%, respectively. In addition to the high-resolution architecture of the dual visual encoder, we also extend the Dense Connector to dynamic high-resolution, specifically using the AnyRes technology from LLaVA-NeXT [25]. For a fair comparison, we provide a baseline for LLaVA-NeXT trained on the same dataset. As shown in Tab. 2, Dense Connector achieves overall improvements compared to the dynamic resolution method LLaVA-NeXT as well.

**Study on Efficient Dense Connector.** To achieve faster inference speed, we investigate an efficient Dense Connector in this study, which can accelerate inference by 3 times. As described in Sec. 3.3, we use a 2D bilinear interpolation function to downsample the visual tokens by a factor of 4, reducing the number of tokens from 576 to 144, which decreases the training time during in the second stage on 8 A100 GPUs from 9 hours to 6.5 hours. With the same configuration of using 144 visual tokens, Dense Connector outperforms the carefully designed efficient connector method Tokenpacker [56] by 0.9%, 1.5%, 2.5% and 1.4% on GQA [58], VQAv2 [59], MMB [64], and MM-Vet [65], respectively.

### 4.3 Main Results

**Comparison with SoTAs in Image Understanding.** In Tab. 4, we scale the LLMs from 2.7B to 70B parameters and compare them with state-of-the-art MLLMs. When considering lightweight models, our Dense Connector surpasses the previous MLLM, TinyLlava [72], achieving a 1.7% enhancement on the MM-Vet benchmark using the same fine-tuning data and foundation model. Furthermore, using same training data and LLM, our Dense Connector outperforms the LLaVA-1.5 Vicuna 13B [16] with substantial gains of 2.1%, 3.7%, and 5.5% on the GQA [58], MMB [64], and MM-Vet [65] benchmarks, respectively. Notably, even with data solely from LLaVA-1.5, our 13B model achieves performance comparable to MGM [18], which is trained on larger datasets, including 1.2M+1.5M data. Moreover, utilizing the advanced open-source LLM Llama3-8B-Instruct, our model significantly surpasses LLaVA-LLama3 [76] with improvements of 5.5%, and 52 on MMB [64], and MME$^p$ [67], respectively, highlighting the contribution of our Dense Connector. By scaling up the LLM to 34B and 70B, Dense Connector achieves further improvements leveraging more powerful language models. The 70B model attains scores of 82.4% on SQA [60] and 79.4% on MMBench [64]. We then increase the resolution using AnyRes technology [25] and fully fine-tuned the LLM. Our 13B model outperforms MGM and LLaVA-NeXT on MMBench [64] and SQA [60], achieving scores of 72.3% and 72.6%. The 34B model achieves scores of 81.2%, 59.2%, and 97.7% on MMBench [64], MM-Vet [65], and LLaVA-Bench-in-the-Wild [16], respectively.

Table 4: Comparisons with State-of-the-Arts. $^*$ indicates the dataset have been used for training, and $^\dagger$ indicates the dataset is not publicly accessible. "PT," "IT," and "Res." denote pre-training data, instruction fine-tuning data, and image resolution, respectively.

| Method | PT+IT | Res. | LLM | SQA$^I$ | MMB | MME$^p$ | MM-Vet | MMMU$^v$ | Math | LLaVA$^W$ | GQA |
|---|---|---|---|---|---|---|---|---|---|---|---|
| MobileVLM V2 [57] | 1.2M+3.6M | 336 | ML-2.7B | 70.0 | 63.2 | 1441 | – | – | – | – | 61.1 |
| TinyLLaVA [72] | 0.5M+0.6M | 384 | Phi2-2.7B | 69.9 | – | – | 32.1 | – | – | 67.9 | 61.3 |
| mPLUG-Owl2 [73] | 348M+1.2M | 448 | Llama2-7B | 68.7 | 64.5 | 1450 | 36.2 | 32.7 | 22.2 | – | 56.1 |
| Qwen-VL-Chat$^\dagger$ [24] | 1.4B+50M | 448 | Qwen-7B | 68.2 | 60.6 | 1488 | – | – | – | – | 57.5* |
| LLaVA-v1.5 [16] | 0.5M+0.6M | 336 | Vicuna-13B | 71.6 | 67.7 | 1531 | 36.1 | 36.4 | 27.6 | 72.5 | 63.3 |
| ShareGPT4V [17] | 1.2M+0.7M | 336 | Vicuna-13B | 71.2 | 68.5 | 1619 | 43.1 | – | – | 79.9 | 64.8 |
| MobileVLM V2 [57] | 1.2M+3.6M | 336 | Vicuna-7B | 74.8 | 70.8 | 1559 | – | – | – | – | 64.6 |
| LLaMA-VID [74] | 0.8M+0.7M | 336 | Vicuna-7B | 70.0 | 66.6 | 1542 | – | – | – | – | 65.0* |
| SPHINX-Plus [75] | 16M | 448 | Llama2-13B | 74.2 | 71.0 | 1458 | 47.9 | – | 36.8 | 71.7 | – |
| LLaVA-LLaMA3 [76] | 0.5M+0.6M | 336 | Llama3-8B | 73.3 | 68.9 | 1506 | – | 36.8 | – | – | 63.5 |
| CuMo [77] | 0.5M+0.6M | 336 | Mistral-7B | 71.7 | 69.6 | 1429 | 34.3 | – | – | 68.8 | 63.2 |
| MM1 [77] | 3B+1.4M | 1344 | MM1-7B | 72.6 | 79.0 | 1529 | 42.1 | 37.0 | 35.9 | 81.5 | – |
| VILA [78] | 50M+1M | 336 | Llama-2-13B | 73.7 | 70.3 | 1570 | 38.8 | – | – | 73.0 | 63.3* |
| Mini-Gemini [18] | 1.2M+1.5M | 336+768 | Vicuna-13B | 72.6 | 68.5 | 1565 | 46.0 | 38.1 | 37.0 | 87.7 | 63.4 |
| LLaVA-NeXT [25] | 0.5M+0.7M | 336$_{AnyRes}$ | Vicuna-13B | 73.6 | 70.0 | 1575 | 48.4 | 36.2 | 35.3 | 87.3 | 65.4 |
| *Scaling to a wider range of parameter sizes (2B → 70B) for LLMs* | | | | | | | | | | | |
| Dense Connector | 0.5M+0.6M | 384 | Phi2-2.7B | 70.3 | 70.5 | 1487 | 33.8 | 36.6 | 28.2 | 65.1 | 61.5 |
| Dense Connector | 0.5M+0.6M | 384 | Vicuna-7B | 70.5 | 68.4 | 1523 | 35.4 | 36.7 | 25.5 | 67.4 | 64.4 |
| Dense Connector | 0.5M+0.6M | 384 | Vicuna-13B | 73.0 | 71.4 | 1569 | 41.6 | 34.3 | 29.6 | 73.6 | **65.4** |
| Dense Connector | 0.5M+0.6M | 384 | Llama3-8B | 75.2 | 74.4 | 1558 | 34.6 | 40.4 | 28.6 | 68.8 | 65.1 |
| Dense Connector | 0.5M+0.6M | 384 | Yi-34B$_{LoRA}$ | 80.5 | 77.7 | 1588 | 41.0 | 47.1 | 33.5 | 75.1 | 63.9 |
| Dense Connector | 0.5M+0.6M | 384 | Llama3-70B$_{LoRA}$ | **82.4** | 79.4 | 1622 | 46.1 | 47.0 | 32.9 | 74.5 | 64.0 |
| Dense Connector | 1.2M+1.5M | 384 | Vicuna-13B | 77.1 | 74.4 | 1579 | 47.8 | 37.2 | 36.5 | 88.9 | 64.6 |
| Dense Connector | 1.2M+1.5M | 384$_{AnyRes}$ | Vicuna-7B | 72.0 | 69.2 | 1535 | 44.4 | 36.4 | 32.7 | 88.8 | 63.9 |
| Dense Connector | 1.2M+1.5M | 384$_{AnyRes}$ | Vicuna-13B | 75.2 | 72.3 | 1573 | 47.0 | 36.8 | 35.5 | 93.2 | 64.3 |
| Dense Connector | 1.2M+1.5M | 384$_{AnyRes}$ | Yi-34B | 78.0 | **81.2** | **1696** | **59.2** | **51.8** | **40.0** | **97.7** | **66.6** |

Table 5: Comparisons with Leading Methods on Zero-shot Video QA Benchmarks. Following FreeVA [54], we specify the GPT-3.5 versions used for evaluation to ensure fairness in performance comparison across different versions. "MAR" denotes the `GPT-3.5-Turbo-0301`, "JUN" denotes the `GPT-3.5-Turbo-0613`, and "JAN" denotes the latest `GPT-3.5-Turbo-0125`.

| Method | LLM Size | GPT-3.5 Version | Train Free | MSVD-QA Acc | MSVD-QA Score | MSRVTT-QA Acc | MSRVTT-QA Score | ActivityNet-QA Acc | ActivityNet-QA Score | Video-ChatGPT CI | DO | CU | TU | CO |
|---|---|---|---|---|---|---|---|---|---|---|---|---|---|---|---|
| FrozenBiLM [79] | 0.9B | MAR | ✗ | 33.8 | – | 16.7 | – | 25.9 | – | – | – | – | – | – |
| Video-LLaMA[52] | 7B | MAR | ✗ | 51.6 | 2.5 | 29.6 | 1.8 | 12.4 | 1.1 | 1.96 | 2.18 | 2.16 | 1.82 | 1.79 |
| LLaMA-Adapter [80] | 7B | MAR | ✗ | – | – | – | – | – | – | 2.03 | 2.32 | 2.30 | 1.98 | 2.15 |
| VideoChat [81] | 7B | MAR | ✗ | 56.3 | 2.8 | 45.0 | 2.5 | 26.5 | 2.2 | 2.23 | 2.50 | 2.53 | 1.94 | 2.24 |
| Video-ChatGPT [51] | 7B | MAR | ✗ | 64.9 | 3.3 | 49.3 | 2.8 | 35.2 | 2.7 | 2.50 | 2.57 | 2.69 | 2.16 | 2.20 |
| VaQuitA [82] | 7B | MAR | ✗ | 74.6 | 3.7 | 68.6 | 3.3 | 48.8 | 3.3 | – | – | – | – | – |
| LLaVA+FreeVA [54] | 7B | MAR | ✓ | 81.5 | 4.0 | 72.9 | 3.5 | 58.3 | 3.5 | 2.88 | 2.52 | 3.25 | 2.32 | 3.07 |
| BT-Adapter [83] | 7B | JUN | ✗ | 67.5 | 3.7 | 57.0 | 3.2 | 45.7 | 3.2 | 2.68 | 2.69 | 3.27 | 2.34 | 2.46 |
| Video-LLaVA [53] | 7B | JUN | ✗ | 70.7 | 3.9 | 59.2 | 3.5 | 45.3 | 3.3 | – | – | – | – | – |
| LLaMA-VID [74] | 13B | JUN | ✗ | 70.0 | 3.7 | 58.9 | 3.3 | 47.5 | 3.3 | 3.07 | 3.05 | 3.60 | 2.58 | 2.63 |
| LLaVA+FreeVA [54] | 13B | JUN | ✓ | 71.8 | 3.8 | 59.2 | 3.3 | 54.5 | 3.5 | 2.90 | 2.52 | 3.26 | 2.32 | 3.07 |
| LLaVA+FreeVA [54] | 13B | JAN | ✓ | 74.4 | 4.1 | 61.1 | 3.6 | 51.6 | 3.5 | 2.88 | 2.52 | 3.25 | 2.34 | 3.05 |
| DC+FreeVA | 7B | JAN | ✓ | 75.0 | 4.1 | 58.4 | 3.5 | 52.2 | 3.5 | 2.80 | 2.51 | 3.17 | 2.22 | 3.05 |
| DC+FreeVA | 13B | JAN | ✓ | 75.1 | 4.1 | 60.8 | 3.5 | 52.6 | 3.5 | 2.85 | 2.53 | 3.23 | 2.29 | 2.96 |
| DC+FreeVA | 34B | JAN | ✓ | 77.4 | 4.2 | 62.1 | 3.6 | 55.8 | 3.6 | 3.00 | 2.53 | 3.25 | 2.65 | 2.92 |

**Comparison with SoTAs in Video Understanding.** Building on the training-free paradigm of FreeVA [54] for image-to-video adaptation, we directly apply our models, originally trained on image-text datasets, to video dialogues. As indicated in Tab. 5, the visual enhancement capabilities of the Dense Connector significantly enhance the video comprehension of our 13B model. This model surpasses the baseline LLaVA-1.5 [16] with FreeVA [54] on MSVD [68] and ActivityNet [69]. By scaling up the LLM to 34B, our model demonstrates a marked improvement in video understanding capabilities, achieving state-of-the-art performance on video evaluation benchmarks with accuracies of 77.4% on MSVD [68], 62.1% on MSR-VTT [70], and 55.8% on ActivityNet [69].

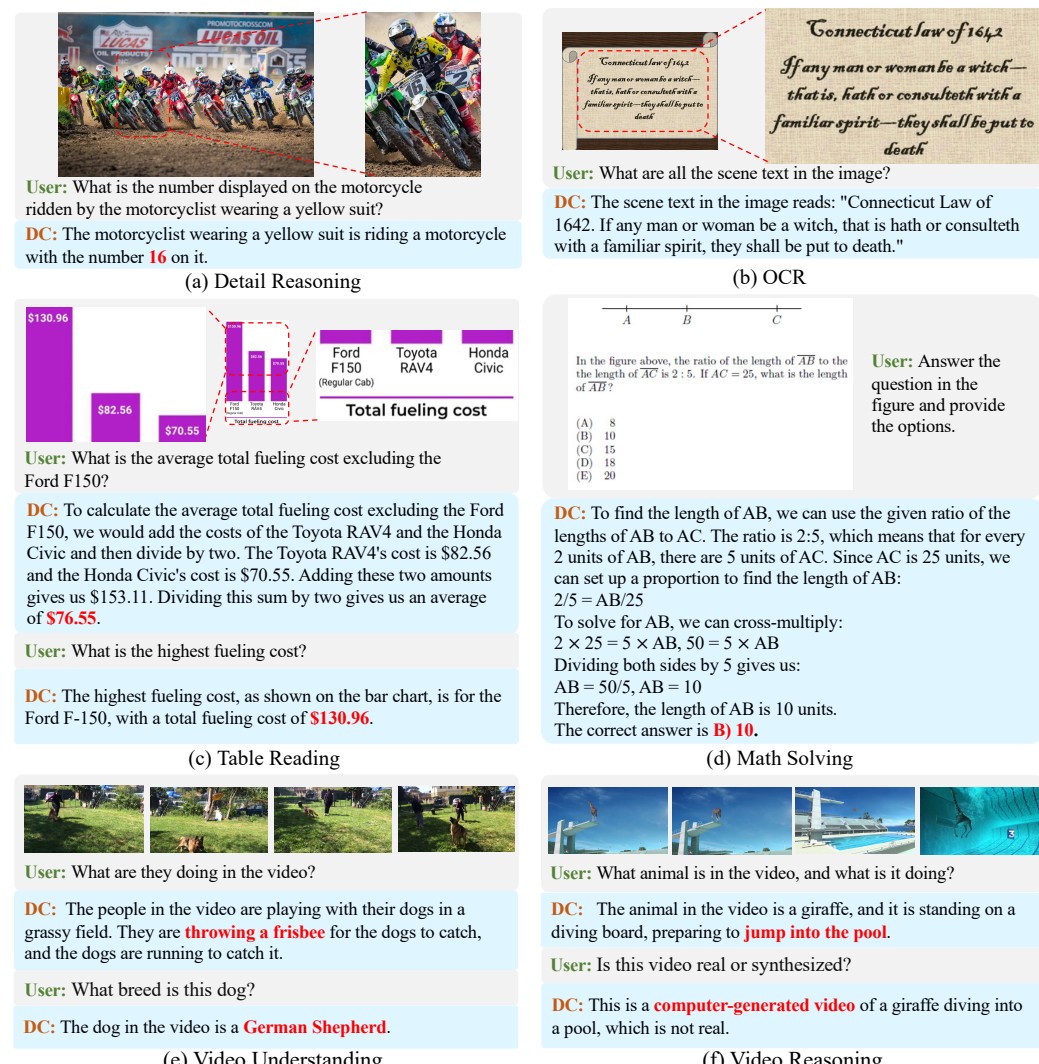

Figure 3: Quantitative Results for Image and Video dialogues. Figures (a) through (d) pertain to image understanding, while figures (e) and (f) relate to video understanding.

**Qualitative Results.** In Fig. 3, we illustrate our model's exceptional visual understanding and text generation capabilities across various scenarios, encompassing both image and video. More qualitative results are provided in Appendix A.5.

## 5 Conclusion and Limitation

In this paper, we introduce the Dense Connector, a novel plug-and-play module that enhances visual perception capabilities of MLLMs by densely integrating multi-layer visual features. We instantiated three types of Dense Connector and validate the efficacy of it across a diverse array of vision encoders, LLMs, and training datasets, demonstrating substantial improvements in performance across multiple evaluation benchmarks. Dense Connector can be easily integrated into existing MLLMs. In this work, we incorporate the Dense Connector into mainstream model LLaVA and high-resolution method Mini-Gemini, demonstrating its versatility and generalization capabilities.

**Limitations:** Our three Dense Connector instantiations do not introduce additional parameters, leaving room for further exploration. We have not yet found an effective method for incorporating additional parameters. Future research will focus on discovering more efficient ways to connect visual and language models for better modality alignment.

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

# A Appendix

## A.1 Further Exploration and Analysis of the Dense Connector

In Sec. 3.2, we provide a detailed discussion of three instantiation methods of the Dense Connector: Sparse Token Integration (STI), Sparse Channel Integration (SCI), and Dense Channel Integration (DCI). Applying our proposed Dense Connector to representative VLMs (*e.g.*, LLaVA [16]) results in notable performance improvements. In this section, building on the STI, SCI, and DCI methods, we explore instantiations of Dense Connectors with additional learnable parameters.

**Sparse Token Integration with 1D Convolutional Downsampling.** The STI instantiation method concatenates visual tokens from various layers, which significantly increases the total number of tokens. To reduce redundancy and computational overhead, we initially employ average pooling to reduce the number of tokens from shallow layer features. In computer vision, using convolution for downsampling is a common practice [84]. Here, we replace average pooling with a single 1D convolutional layer $Conv_{1D}(\cdot)$, to downsample the shallow layer tokens. We set both the kernel size and stride to 8, maintaining consistency with the average pooling as detailed in Sec. 4.1. The formula is as follows:

$$e_v = MLP(Concatenate([Conv_{1D}(V_{l_1}), ..., Conv_{1D}(V_{l_K}), V_L], dim = token)). \tag{5}$$

**Sparse Channel Integration with 2D Convolutional Modelling.** The SCI method utilizes the projector as both a feature fusion tool and a modality mapper, transforming multi-layer visual features concatenated along the channel dimension into the input feature space of the LLMs. By straightforwardly concatenating multi-layer visual features, the SCI method achieves significant performance enhancements for VLMs with minimal computational overhead. As a central element of the Dense Connector, we aim to enhance local perception abilities by processing visual features through a 3×3 2D convolution $Conv_{2D}$, with shared weights prior to feature concatenation:

$$e_v = MLP(Concatenate([Conv_{2D}(V_{l_1}), ..., Conv_{2D}(V_{l_K}), Conv_{2D}(V_L)], dim = channel)). \tag{6}$$

**Dense Channel Integration with Linear Layer.** The DCI method initially groups visual features and then aggregates them, enabling the fusion of adjacent visual features without expanding the dimensionality. This technique mitigates the issues of feature redundancy and excessive channel dimensions that arise in the SCI method from incorporating too many layers. To enhance the integration of features from different visual layers across groups, each layer of visual features is processed through a linear layer with shared weights. The formula is as follows:

$$GV_g = \frac{1}{M} \sum_{i=(g-1)M+1}^{gM} Linear(Ln(V_i)), \quad 1 \leq g \leq G. \tag{7}$$

$$e_v = MLP(Concatenate([GV_1, ..., GV_G, V_L], dim = channel)), \tag{8}$$

where $Ln(\cdot)$ and $Linear(\cdot)$ denotes Layer Normalization and Linear layer, respectively.

**Experimental Results and Analysis.** We present the detailed results of these described instantiations in Tab. 6. 1) For the STI method, using features from the 8th, 16th, and 24th layers, average pooling demonstrates superior performance compared to 1D convolutional downsampling, especially on the GQA [58] and MMBench [64] benchmarks. 2) In the SCI method, applying 2D convolution to enhance local feature modeling with offline ViT features from the 8th, 16th, and 24th layers achieves comparable performance to methods without 2D convolution. 3) Furthermore, incorporating additional aggregation information, such as a linear layer, into the DCI method does not lead to improved outcomes.

In summary, our attempts to introduce additional parameterized modules did not yield the anticipated improvements. We hypothesize that since the connector is randomly initialized, maintaining ease of convergence is crucial for effectively training the connector to align visual and language models under

Table 6: Additional studies on the Dense Connector.

| Architecture | Layer Index | GQA | VQA$^{v2}$ | SQA$^I$ | MMB | MMVet |
|---|---|---|---|---|---|---|
| LLaVA [16] | 24 | 62.0 | 78.5 | 66.8 | 64.3 | 31.1 |
| + *STI* | 8,16,20,24 | 63.0 | 79.1 | 68.0 | **67.6** | 30.8 |
| + *STI* | 8,16,24 | **63.3** | **79.1** | **68.0** | 67.2 | **30.9** |
| + *STI w/* 1D Conv | 8,16,24 | 62.6 | 78.9 | 67.4 | 65.0 | 30.6 |
| + *SCI* | 8,16,24 | **63.7** | **79.2** | **68.9** | **66.2** | **32.2** |
| + *SCI* | 4,8,16,24 | 62.9 | 78.9 | 68.3 | 65.2 | 30.2 |
| + *SCI* | 8,12,16,24 | 63.5 | 79.0 | 67.4 | 65.6 | 31.7 |
| + *SCI* | 8,16,20,24 | 63.6 | 79.2 | 67.0 | 65.8 | 31.9 |
| + *SCI w/* 2D Conv | 8,16,24 | 63.6 | 79.0 | 68.1 | 66.0 | 32.2 |
| + *DCI* | (1-12),(13-24) | **63.8** | **79.5** | **69.5** | **66.8** | 32.7 |
| + *DCI w/* Linear | (1-12),(13-24) | 63.7 | 79.4 | 68.5 | 66.4 | **32.8** |

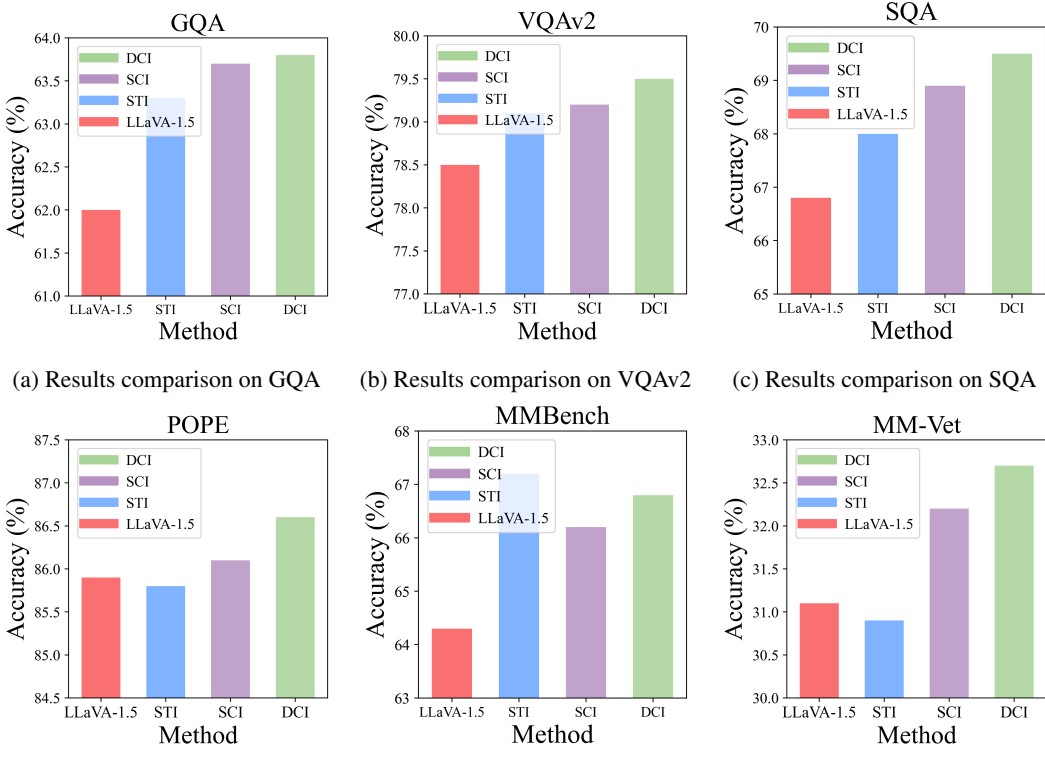

(a) Results comparison on GQA    (b) Results comparison on VQAv2    (c) Results comparison on SQA

(d) Results comparison on POPE    (e) Results comparison on MMB    (f) Results comparison on MM-Vet

Figure 4: Comparison of three instantiations of Dense Connector with LLaVA-1.5. STI stands for Sparse Token Integration, SCI for Sparse Channel Integration and DCI for Dense Channel Integration.

limited training data. In current MLLMs, the MLP structure of LLaVA [16] efficiently facilitates convergence between visual and language components. However, adding extra parameters may disrupt this inherent characteristic, leading to diminished performance. Future research will explore more complex and effective implementations of the Dense Connector.

## A.2 Visualized Analysis

In Appendix A.1, we further explored more complex Dense Connector structures. However, we found that non-parameterized methods yielded better average results. Therefore, we also visualized the comparison of three main instantiations of the Dense Connector with LLaVA-1.5 in Fig. 4, clearly demonstrating DCI's superior performance across different benchmarks.

Table 7: More comprehensive evaluation results of the Dense Connector. "PT", "IT", and "Res." denote pre-training data, instruction fine-tuning data, and image resolution, respectively. ‡ indicates the pre-training data (0.5M) from LLaVA-1.5 [16] and instruction data (1.5M) from Mini-Gemini [18].

| Method | PT+IT | Res. | LLM | GQA | $VQA^T$ | $SQA^I$ | MMB | $MME^p$ | $MMMU^v$ | Math | MMV | $LLaVA^W$ |
|---|---|---|---|---|---|---|---|---|---|---|---|---|
| Dense Connector | 0.5M+0.6M | 384 | Phi2-2.7B | 61.5 | 55.8 | 70.3 | 70.5 | 1487 | 36.6 | 28.2 | 33.8 | 65.1 |
| Dense Connector | 0.5M+0.6M | 384 | Vicuna-7B | 64.4 | 62.6 | 70.5 | 68.4 | 1523 | 36.7 | 25.5 | 35.4 | 67.4 |
| Dense Connector | 0.5M+0.6M | 384 | Llama3-8B | 65.1 | 62.2 | 75.2 | 74.4 | 1558 | 40.4 | 28.6 | 34.6 | 68.8 |
| Dense Connector | 0.5M+0.6M | 384 | Vicuna-13B | **65.4** | 64.7 | 73.0 | 71.4 | 1569 | 34.3 | 29.6 | 41.6 | 73.6 |
| Dense Connector | 1.2M+1.5M | 384 | Vicuna-13B | 64.6 | 65.0 | 77.1 | 74.4 | 1579 | 37.2 | 36.5 | 47.8 | 88.9 |
| Dense Connector | 0.5M+0.6M | 384 | Yi-34B$_{LoRA}$ | 63.9 | 66.7 | 80.5 | 77.7 | 1588 | **47.1** | 33.5 | 41.0 | 75.1 |
| Dense Connector | 0.5M+0.6M | 384 | Llama3-70B$_{LoRA}$ | 64.0 | 66.0 | **82.4** | 79.4 | 1622 | 47.0 | 32.9 | 46.1 | 74.5 |
| Dense Connector | 0.5M+1.5M‡ | 384 | Llama3-70B$_{LoRA}$ | 64.1 | **68.3** | 74.5 | **80.2** | **1649** | 43.6 | **40.7** | **53.3** | **97.8** |

Table 8: Ablation study on fine-tuning Vision Transformer. In this table, all the results are conducted using LLaVA 1.5 data, comprising 558K pre-training data and 665K instruction-tuning data.

| Method | VE | FT-ViT | Res. | LLM | GQA | $SQA^I$ | $VQA^T$ | MMB | MMV | $MMMU^v$ | Math |
|---|---|---|---|---|---|---|---|---|---|---|---|
| DC (*w/* LLaVA) | CLIP-L | ✗ | 336 | Vicuna-7B | 63.8 | 69.5 | 59.2 | 66.8 | 32.7 | 34.8 | 26.9 |
| DC (*w/* LLaVA) | CLIP-L | ✓ | 336 | Vicuna-7B | 63.7 | 67.4 | 60.2 | 68.6 | 34.4 | 35.4 | 26.0 |
| DC (*w/* LLaVA) | Siglip-SO | ✗ | 384 | Vicuna-7B | 64.2 | 70.5 | 62.6 | 68.4 | 35.4 | 36.7 | 25.5 |
| DC (*w/* LLaVA) | Siglip-SO | ✓ | 384 | Vicuna-7B | 65.0 | 71.6 | 63.4 | 69.3 | 35.8 | 35.2 | 27.0 |

## A.3 Model Zoo

To probe the upper limits of MLLMs, we attempt to utilize a larger dataset to fine-tune our model based on Llama3-70B-Instruct. Given the limitations of computational resources, we commence with the Dense Connector pre-trained on LLaVA's initial stage data [16] and subsequently refine it using the instructional dataset from Mini-Gemini [18]. This model achieves accuracy of 40.7% on the MathVista [63] and 53.3% on MM-Vet [65], as illustrated in Tab. 7. Remarkably, our 70B model exhibits performance that is on par with GPT-4V on the LLaVA-Bench-in-the-Wild evaluation [16]. Specifically, our model achieves 97.8%, closely trailing GPT-4V's 98% [85]. However, it is worth noting that the full potential of our 70B model remains untapped due to the lack of initial stage pre-training data and the inherent constraints of LoRA fine-tuning [86].

## A.4 Further Exploration in Training Visual Encoder

In the experiments above, we used frozen multi-layer features from a pre-trained ViT to enhance the model's visual perception capabilities. Recently, however, there has been a growing trend of fine-tuning the entire model, including ViT, during the instruction-tuning phase to achieve better results. Following this trend, we present additional results in Tab. 8 on the performance impact of fine-tuning the ViT within the multi-layer visual connector. Specifically, in the first stage, we maintain the original training strategy. In the second stage, we fine-tune the ViT with a smaller learning rate of 2e-6, keeping other hyperparameters unchanged. Tab. 8 shows that fine-tuning ViT yields performance gains on benchmarks with a higher reliance on visual information, such as MMBench [64] and TextVQA [64]. For instance, fine-tuning CLIP-Large encoder resulted in improvements of 1.8% and 1.0% on MMBench and TextVQA, respectively. Conversely, on benchmarks that are less visually dependent, such as SQA [60], performance exhibited a slight decline.

## A.5 More Qualitative Results

From Fig. 6 to Fig. 13, we present additional multimodal dialogue illustrations with images and videos. Fine-tuned on image-text datasets, Dense Connector achieves effective alignment between visual and linguistic modalities, enhancing the model's visual perception capabilities. Qualitative evaluations showcases that our model can adeptly handle a broad spectrum of challenges across various domains, including mathematics, flowcharts, and daily scenarios. Additionally, the incorporation of the FreeVA method [54] equips our model with video understanding capabilities, enabling our model to address dynamic temporal challenges. However, due to the lack of training on video data, our model may occasionally misinterpret images as videos. For instance, as shown in Fig. 12, the model mistakenly interprets videos as images, incorrectly responding with 'The image' instead of 'The video'.

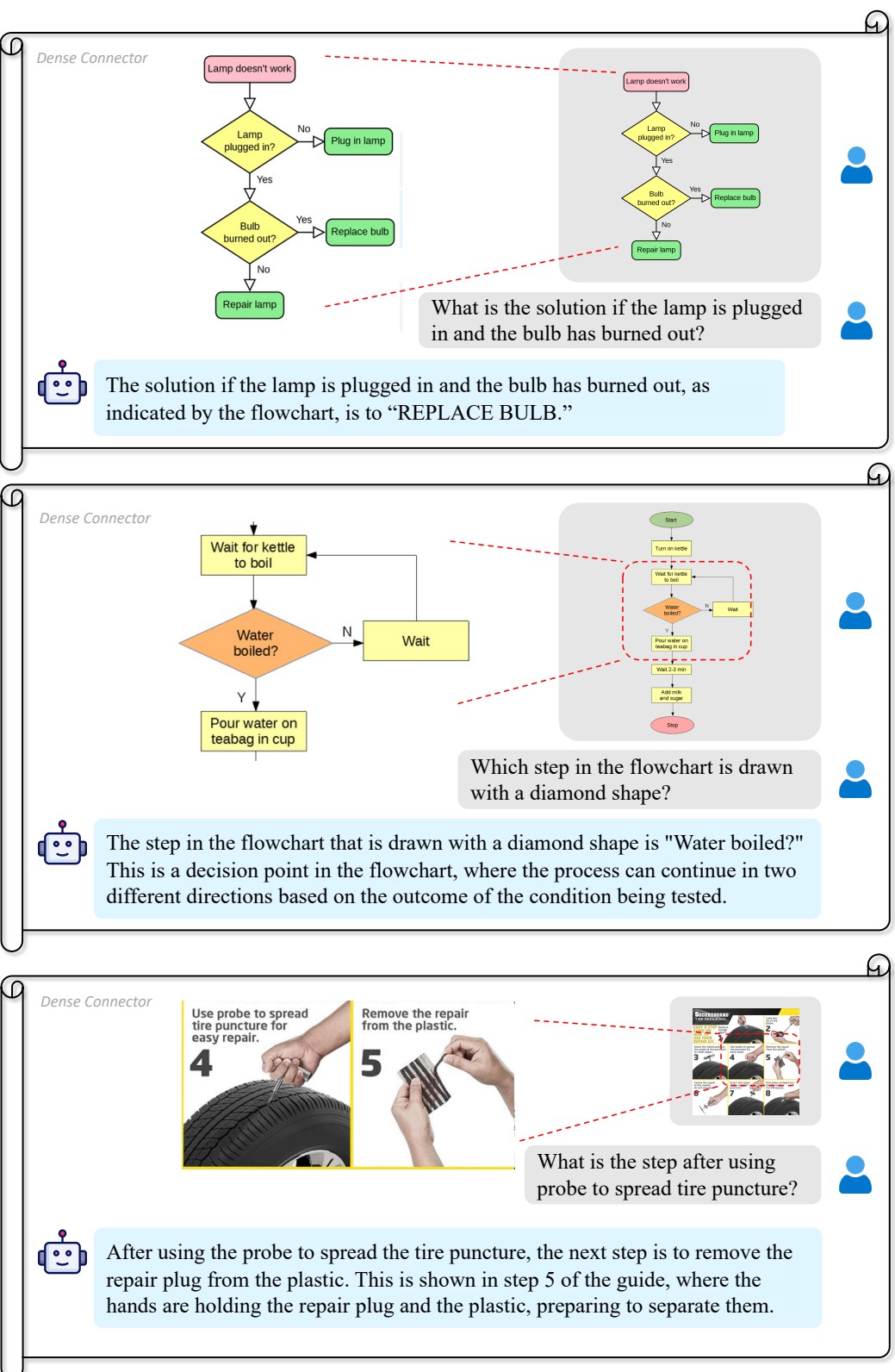

Figure 5: Qualitative results of the flowchart understanding.

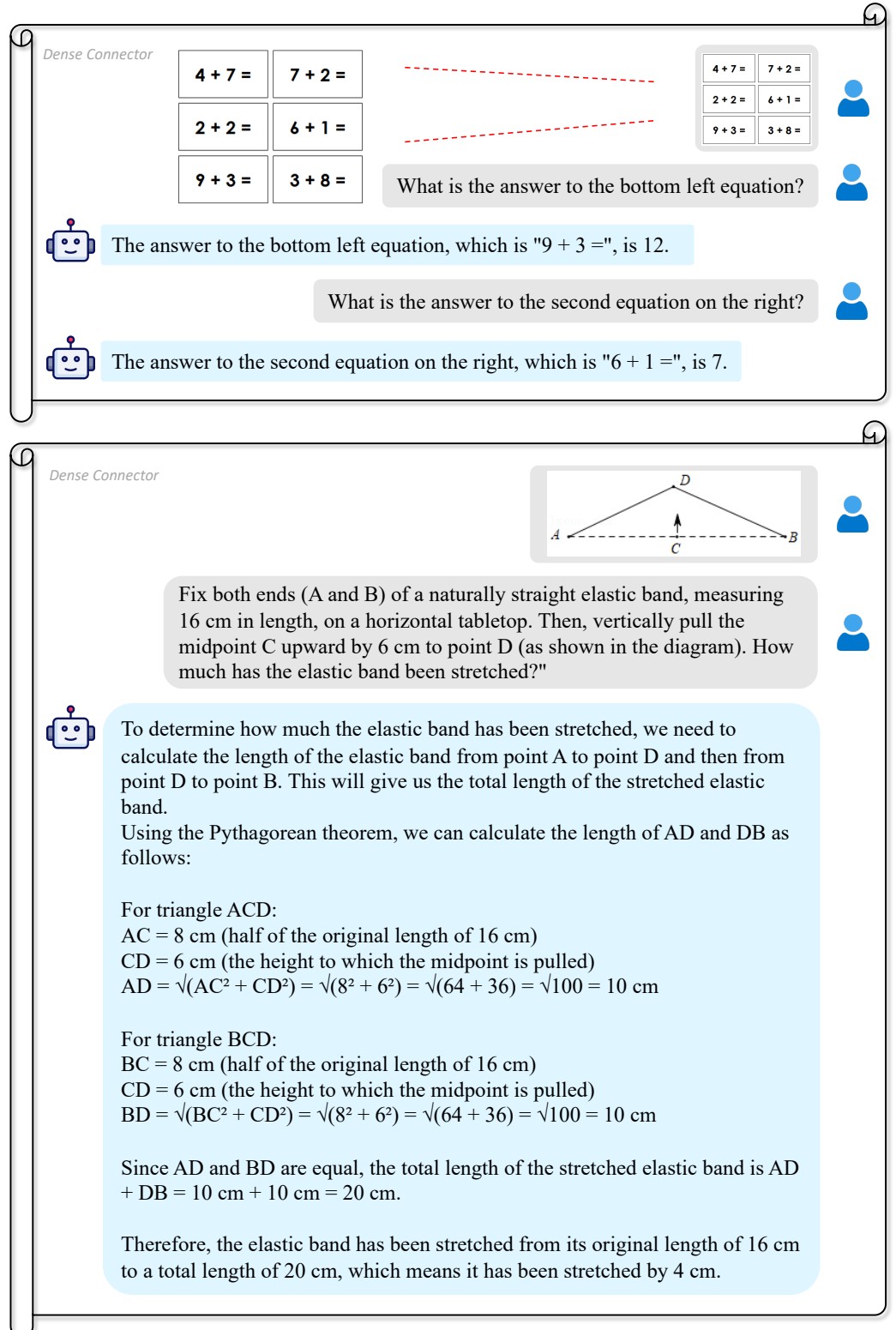

Figure 6: Qualitative results of the math solving skills.

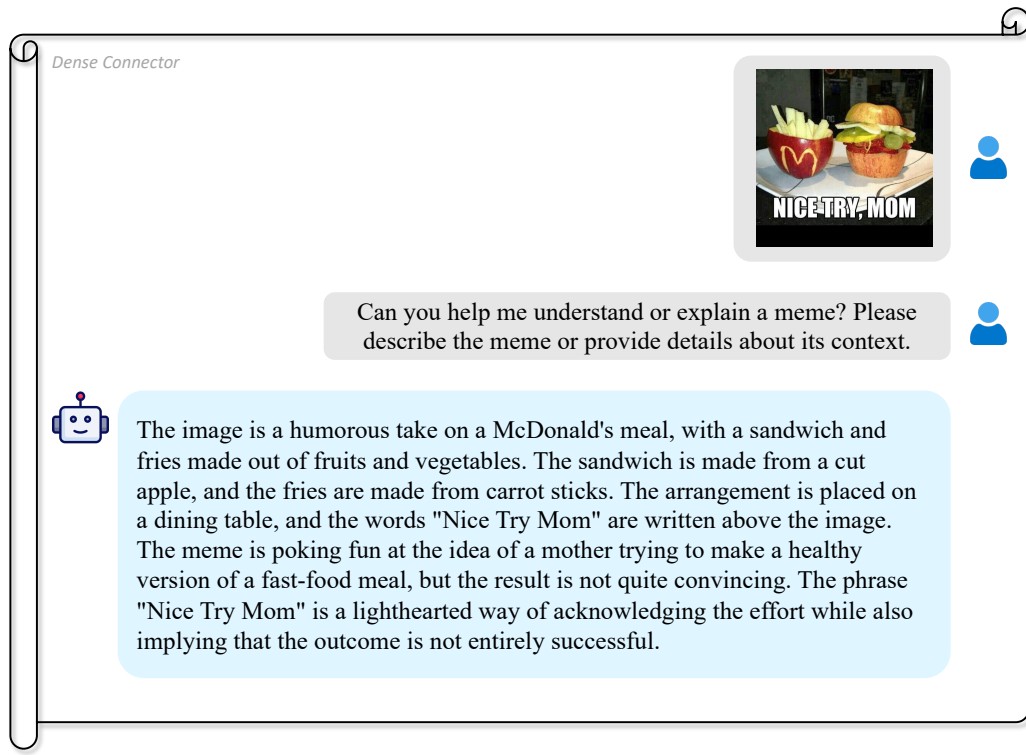

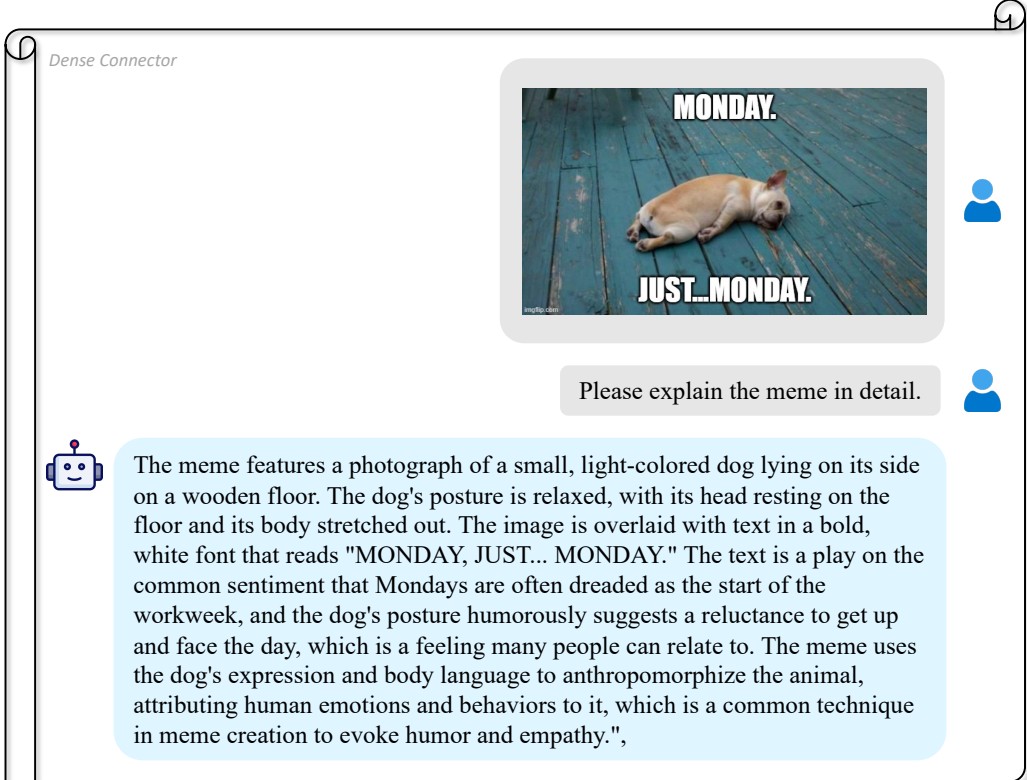

Figure 7: Qualitative results of the meme understanding.

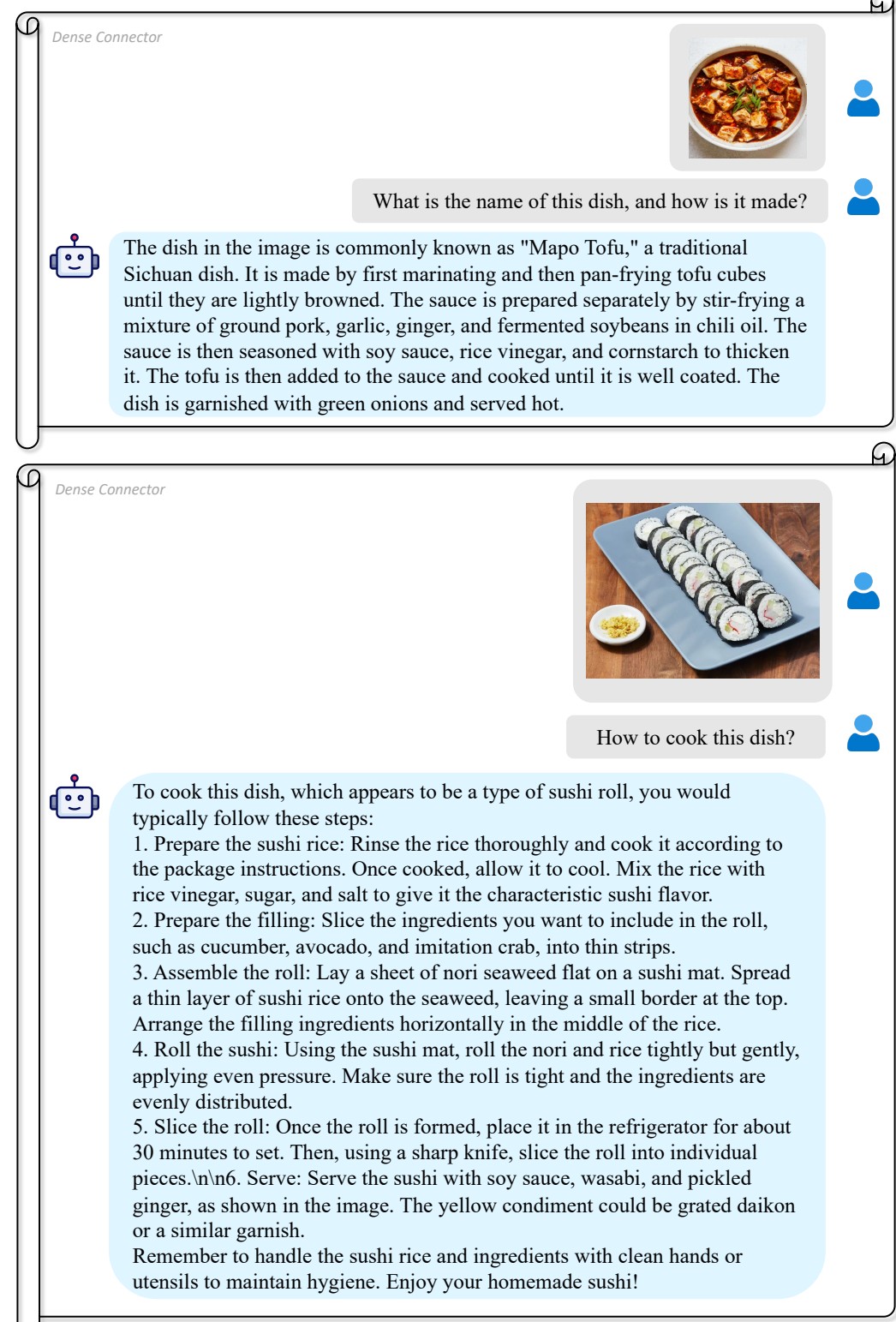

Figure 8: Qualitative results of the delicacies understanding.

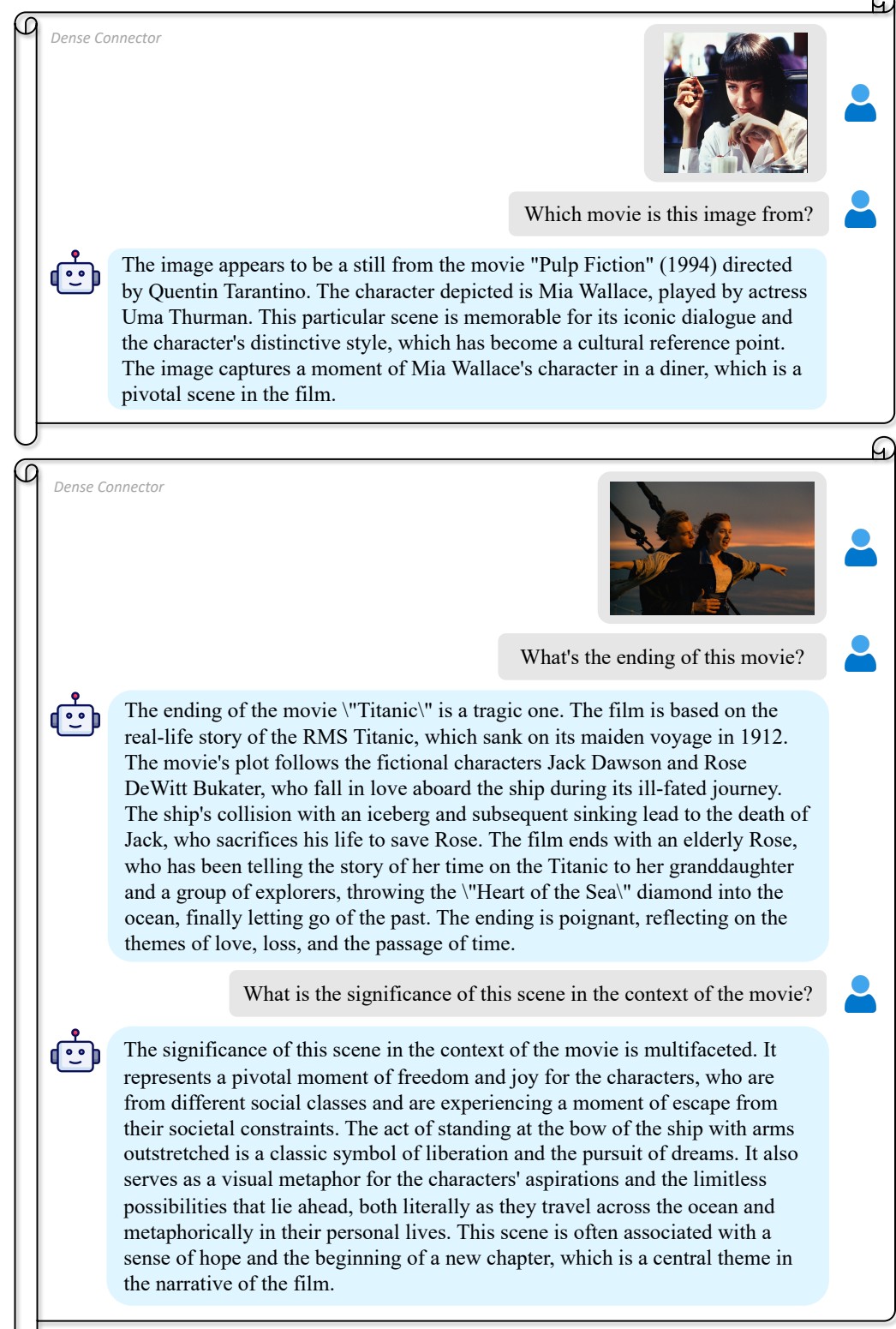

Figure 9: Qualitative results of the movie analysis.

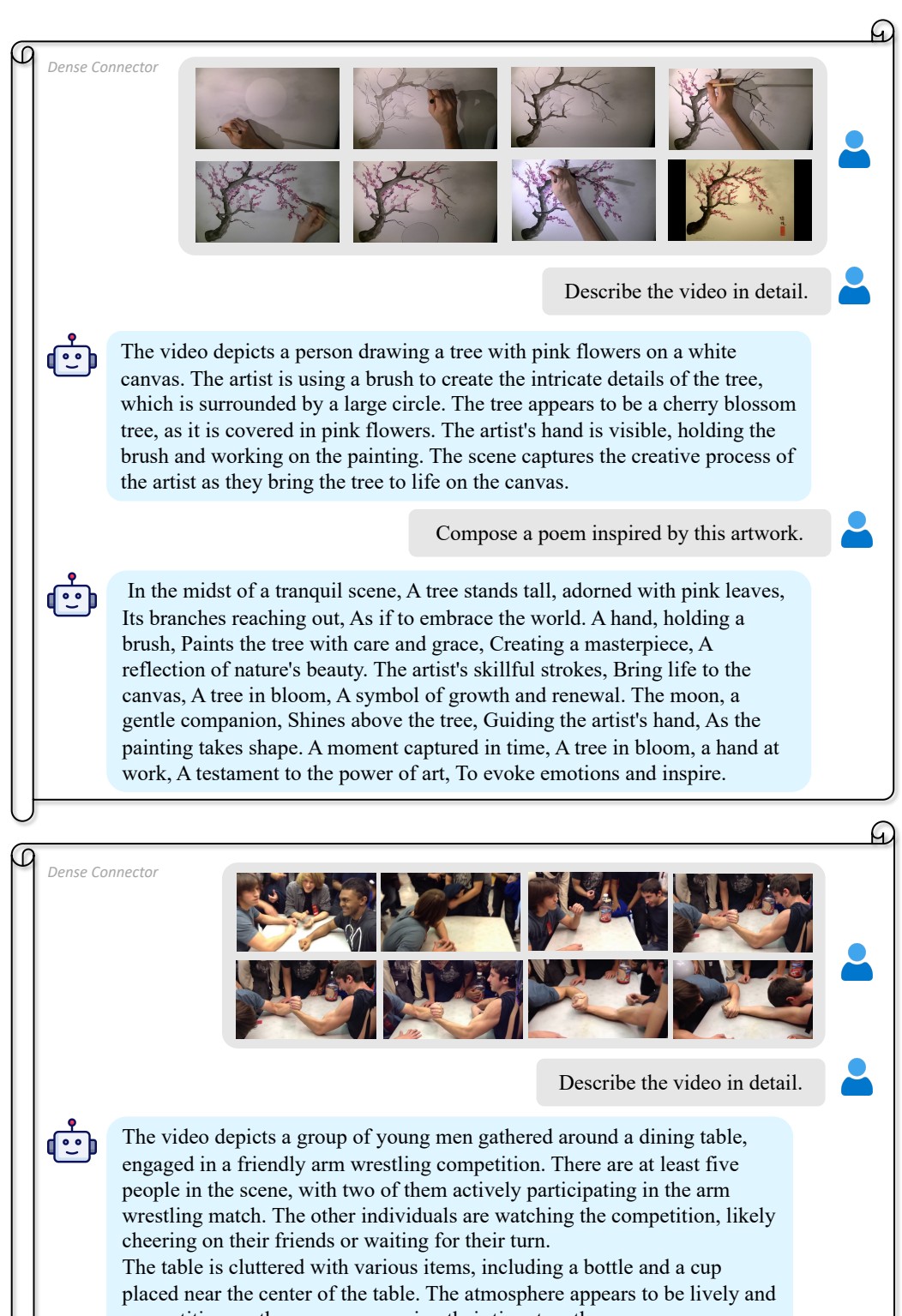

Figure 10: Qualitative results of the video understanding.

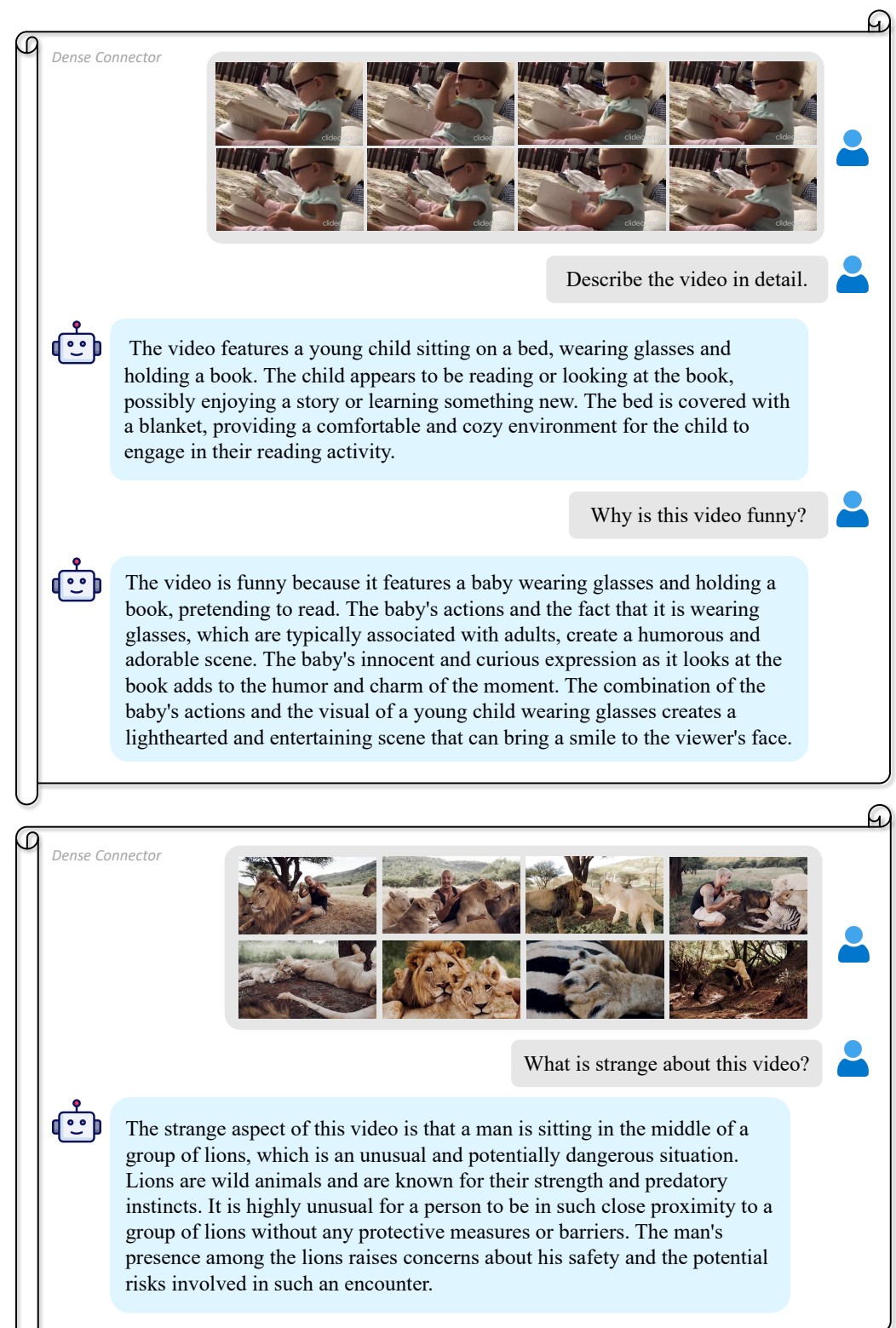

Figure 11: Qualitative results of the video understanding.

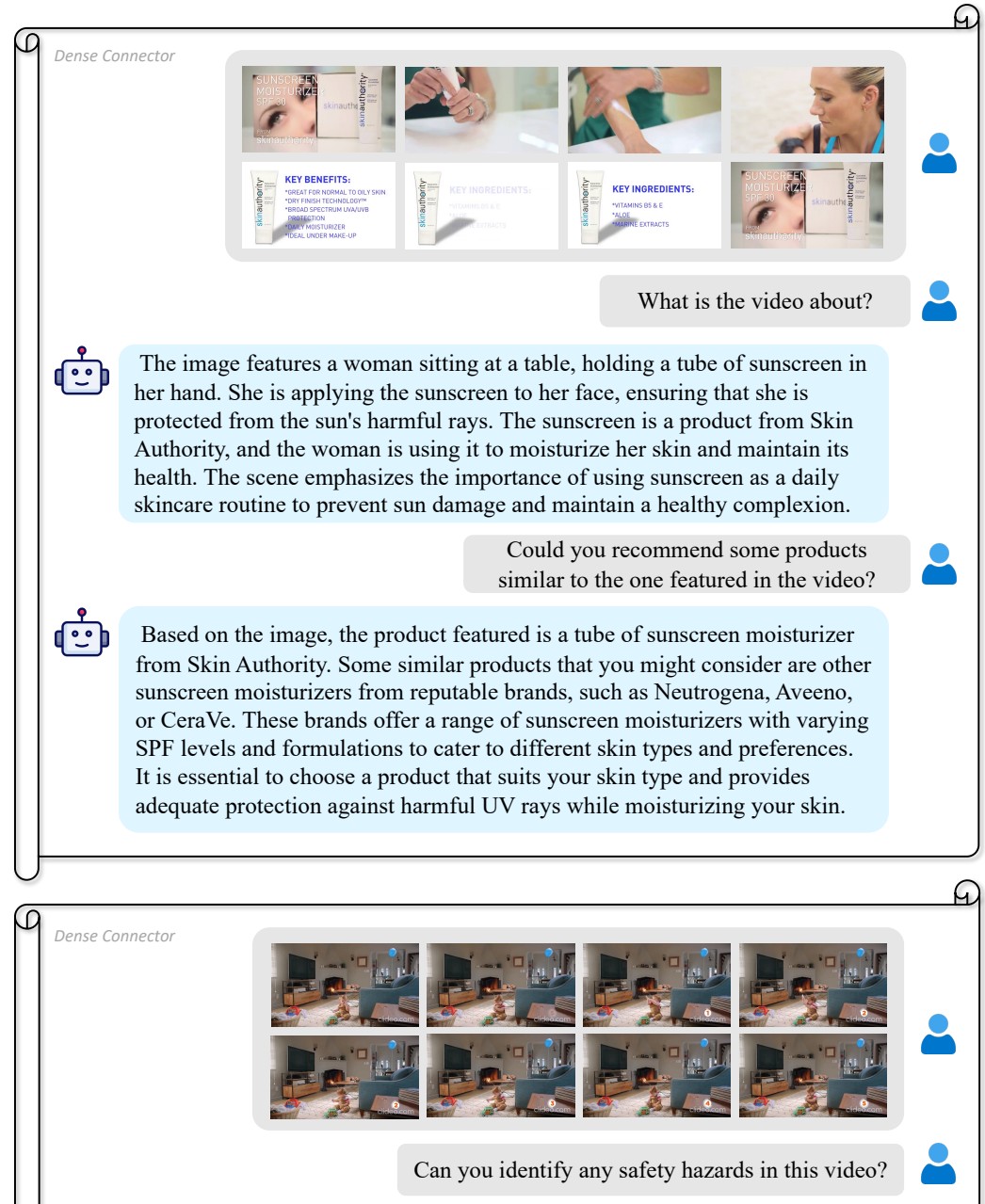

Figure 12: Qualitative results of the video understanding.

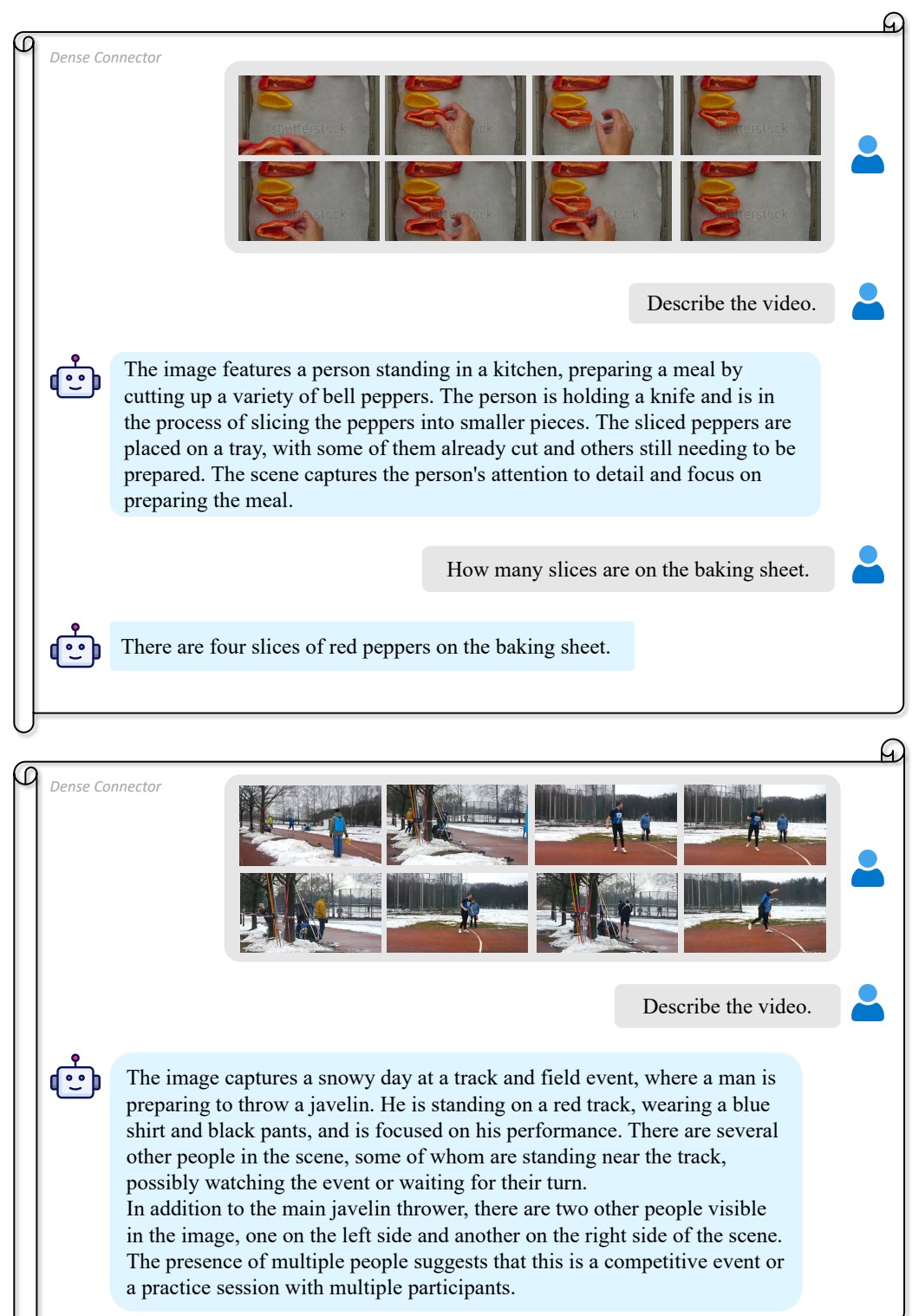

Figure 13: Qualitative results of the video understanding.

