# OpenReview forum: "Dense Connector for MLLMs"
_NeurIPS.cc/2024/Conference — NeurIPS 2024 poster_

### Official Review · Reviewer_qnn9 · 2024-07-12

**Soundness:** 3
**Presentation:** 3
**Contribution:** 3
**Rating:** 6
**Confidence:** 4

**Summary:**

The document presents a novel approach called the Dense Connector (DC), which is a simple and effective plug-and-play vision-language connector that enhances existing Multimodal Large Language Models (MLLMs) by leveraging multi-layer visual features from the frozen visual encoder.The authors propose three intuitive instantiations of the Dense Connector.The authors demonstrate the versatility and scalability of their approach across various visual encoders, image resolutions, training dataset scales, LLM sizes, and MLLM architectures, achieving state-of-the-art performance on 19 image and video benchmarks.

**Strengths:**

1. the dense connector is easily plugged into mainstream MLLM architecture, and could enhance visual representation of existing MLLM with little additional computation.
2. authors demonstrate the compatibility and scalability of proposed approach across visual encoders, image resolutions, scales of training datasets, sizes of LLMs and even for video understanding tasks.

**Weaknesses:**

1. authors do experiments on several MLLM models, however don't summarize the compatible scope of models architectures.

**Questions:**

I am curious whether the DC is compatible with BLIP2-like architectures which have cross-attention visual resamplers?
Could authors give generalization of the compatible types of MLLM architecture?

**Limitations:**

authors adequately addressed the limitations

---

> ### Author Rebuttal · Authors · 2024-08-05
>
> **Q1.Can DC be compatible with the visual resampler architecture?**
>
> This is a very worthwhile research question. Dense Connector (DC) is compatible with models similar to BLIP-2, which have visual resampler or Qformer structures.
>
> Notably, models like BLIP-2 use visual resampler to obtain learnable queries as final visual representations, **which still need MLPs for converting visual queries into the input space of the language model** in structures like BLIP-2. We can replace these MLPs with DC to enhance visual perception. Thus, the DC can be used wherever a vision encoder and a connector are present.
>
> Specifically, the visual resampler compresses high-level features into learnable queries. We then use an interpolation function to downsample tokens of different layer features to align the number of queries. DC is then used to transform these features from different layers into the input space of the language model. The table below shows our experimental results: using the visual resampler architecture, DC improved the average performance by **2\%**.
>
> | Model | GQA | VQAv2 | TextVQA | SQA | MMbench | MM-Vet | AVG |
> | --- | --- | --- | --- | --- | --- | --- | --- |
> | visual resampler w/o DC | 56.2 | 72.7 | 52.0 | 67.3 | 60.5 | 26.0 | 55.8 |
> | visual resampler w/ DC | 58.2 | 73.9 | 53.9 | 68.8 | 62.8 | 29.3 | 57.8 (**+2\%**) |
>
> It is important to note, as referenced in [1], that the larger number of parameters in the visual resampler leads to poorer convergence. As a result, when trained on the same data, the performance of the visual resampler is suboptimal. Nonetheless, the table above demonstrates that **DC is compatible with the visual resampler** and enhance its performance.
>
> [1] Liu, Haotian, et al. "Improved baselines with visual instruction tuning." *Proceedings of the IEEE/CVF Conference on Computer Vision and Pattern Recognition*. 2024.
>
> **Q2.Summarize the model architectures compatible with Dense Connector**
>
> Thank you for pointing this out! We agree that summarizing the compatible scope of model architectures for DC would enhance the clarity and comprehensiveness of this paper.
>
> DC can be widely applied to current MLLM architectures. In current MLLMs, **any model utilizing a vision encoder** can integrate the DC. The reason is that, regardless of whether the models are based on MLPs (LLaVA) or Qformers (BLIP-2), they require linear layers to convert visual features into the language space. Specifically, LLaVA employs two linear layers for high-level feature transformation, while BLIP-2 initially uses Qformer to obtain learnable queries, followed by a single linear layer for feature conversion. **This linear layer is where the DC comes into play.** Therefore, any model using a vision encoder can be compatible with the DC.
>
> In summary, in this paper, we applied DC to a wide range of different architectures, including LLaVA-1.5, Mini-Gemini, **LLaVA-NeXT (please refer to reviewer sj7M's Q1)**, and the Visual Resampler architecture. We will include these in the revision.

---

> > ### Comment · Reviewer_qnn9 · 2024-08-13
> >
> > Thank you for the author's feedback.
> > I want to know, what are the models with visual resampler used in your experimental results above?

---

> ### Author Response · Authors · 2024-08-13
> **Response to reviewer qnn9**
>
> Thank you for the reviewer's comments.
>
> We used a visual resampler based on **Flamingo** [1], with 64 query tokens, 6 layers, a hidden size of 1024, and 16 heads.
> In the first stage, we fine-tune the visual resampler and the linear layer that follows it, while freezing the ViT and LLM. In the second stage, we fine-tune the visual resampler, the linear layer, and the LLM, keeping only the ViT frozen.
>
> Thank you very much for your time and comments! Please let us know if there are any further questions that need clarification.
>
> [1] Alayrac, Jean-Baptiste, et al. "Flamingo: a visual language model for few-shot learning." Advances in neural information processing systems 35 (2022): 23716-23736.

---

### Official Review · Reviewer_2pJ5 · 2024-07-12

**Soundness:** 2
**Presentation:** 2
**Contribution:** 2
**Rating:** 5
**Confidence:** 5

**Summary:**

This paper endeavors to delve into the visual representations in MLLMs, introducing a module, plug-and-play component named as Dense Connector (DC). This  DC is designed to enhance visual representation. To this end, three  instantiations are presented, including Sparse Token Integration(STI), Sparse Channel Integration(SCI), and Dense Channel Integration(DCI).  Thorough experimental evaluations are meticulously carried out to validate the compatibility and scalability, thereby highlighting their potential in enhancing MLLM performance.

**Strengths:**

1. A plug-and-play module can be applyed into various MLLMs.

2. The experiments, including different  training datasets, diverse MLLMs, are conduted to demonstrate the performance of DC. The video benchmarks are also adopted as an extension.

**Weaknesses:**

1. The insight of enhancing visual representations from multi-level features of vision encoder have been explored in previous works, like [1]. The work [1]  has provided the simliar evidences from multi-level features, and more comprehensive analysis based on a amout of experimetal results.
Would authors provides any possible fair comparsions with  the work[1] (Only CLIP vision encoder)?

   [1] Jiang et al.,  From CLIP to DINO: Visual Encoders Shout in Multi-modal Large Language Models, Arxiv 2023.

2. While this work outlines three architectural configurations for the Dense Connector (DC),  it primarily employs a single variant – the Dense Channel Integration (DCI). It is important to note that the DC module serves more as an empirical trick rather than a theoretical methodology.

3.  I think the performance gain is limited, especially on high-resolution MGM baseline in Table2. The  scalability of DC may be limited.
 To comprehensively assess its effectiveness, I encourage the authors to extend their comparisons utilizing a Vicuna-13B model based on the MGM framework, incorporating additional benchmark datasets such as VQAv2, MME, and POPE. It is speculated that the incremental performance gain might become even less pronounced when employing larger language models (LLMs),

**Questions:**

It would be beneficial to include a line chart that incorporates a broader range of combinations, facilitating a clearer visual comparison and enabling the selection of groups more evidently.

**Limitations:**

Suggestions: The presented DC can be furthre to combine multiple vision encoder, like DINOv2.

---

> ### Author Rebuttal · Authors · 2024-08-05
>
> **Q1. More comparisons with COMM [1]**
>
> [1] combines visual features from all layers by simply adding them together, which can lead to **information loss**. Additionally, ViT's low, middle, and high layers contain different information. Therefore, COMM's approach of adding them all together **lacks prior knowledge**.
>
> In contrast, DC groups multi-layer visual features, **providing prior knowledge during integration.** Additionally, DC concatenates multi-level features along the channel dimension, effectively utilizing the dimensional transformation characteristics of MLPs connectors. **It achieves feature fusion and transformation without additional modules.** The tables below show the performance comparison between [1] and DC.
>
> | | POPE | VQAv2 | MME |
> |-|-|-|-|
> |COMM|83.6|70.1|1235|
> |DC|86.6|79.5|1523|
>
> > Would authors provides any possible fair comparisons with the work [1] (Only CLIP vision encoder)?
>
> To ensure a fair comparison, we fine-tuned the model using **LLaVA-1.5 data and a single ViT CLIP-L/336px**. For COMM, we used LLN-Layerscale(all) [1] for integrating visual features.
>
> ||TextVQA|MME|SQA|POPE|LLaVA-Bench|AVG|
> |-|-|-|-|-|-|-|
> |LLaVA (baseline)|58.2|1511|66.8|85.9|65.4|70.4|
> |LLaVA+COMM|58.0|1493|67.4|86.0|60.5|69.3(**-1.1%**)|
> |LLaVA+DC |59.2|1511|69.5|86.6|66.1|71.4(**+1\%**)|
>
> The results above indicate that DC is a **more effective** method, achieving an average performance improvement of **2.1%** compared with [1].
>
> [1] Jiang et al., From CLIP to DINO: Visual Encoders Shout in Multi-modal Large Language Models, Arxiv 2023.
>
> **Q2.Statement about DC**
>
> > While this work outlines three architectural configurations for the DC, it primarily employs a single variant – the Dense Channel Integration (DCI).
>
> Current MLLMs research primarily focuses on high resolution and data, with less attention to multi-layer features. Previous work [1] has simply added visual features together, lacking further discussion. Our paper fills this gap by being the first to extensively discuss the fusion of multi-layer visual signals along the token (STI) and channel (SCI, DCI) dimensions.
>
> We explored three instantiation methods, all enhancing MLLM performance across various settings. Among them, DCI achieved the best performance, so we primarily adopted it for scaling-up experiments.
>
> > It is important to note that the DC module serves more as an empirical trick rather than a theoretical methodology.
>
> DC was designed to enhance MLLM performance from a visual perspective. Inspired by DenseNet and FPN, we explored the use of multi-layer visual features in MLLM and sought effective model designs. Thus, we developed DC, which utilizes multi-layer features in a way **distinct from previous works**. This module is tailored to MLLM, utilizing the dimension transformation of connectors to fuse multi-layer visual features.
>
> Importantly, we believe this work **benefits the community**. First, DC outperforms previous connectors, demonstrating **applicability across architectures** (LLaVA-1.5, MGM, LLaVA-NeXT and Qformer). Second, we address the gap in integrating multi-layer visual features from the token and channel dimensions. Finally, we hope this paper will attract more attention to the visual modality in MLLM.
>
> **Q3.Performance of DC on MGM with Vicuna-13B**
>
> > I think the performance gain is limited, especially on high-resolution MGM.
>
> We believe the improvement brought by DC is significant. DC improved performance on GQA, SQA and MMB by **1.8, 2.7 and 2.5**, respectively. In comparison, a more powerful ViT (CLIP-L->SigLIP) only improved GQA, SQA, MMB by **0.4, 1.0 and 1.6**, and a stronger LLM (7B->13B) only improved by **1.2, 2.5 and 3.0**.
>
> Based on reviewer sj7M's suggestion, we **extended DC to other high-resolution methods LLaVA-NeXT**, achieving performance improvements. Please refer to reviewer sj7M's Q1.
>
> > I encourage the authors to extend their comparisons utilizing a Vicuna-13B model based on the MGM framework.
>
> We extended the MGM experiments to Vicuna-13B with additional benchmark results. The table shows that **DC can enhance MGM-13B's performance across various benchmarks.**
>
> ||VQAv2|MMBench|POPE|MME|LLaVA-Bench|MMVet|
> |-|-|-|-|-|-|-|
> |MGM|80.8|68.5|85.1|1565/322|87.5|46.0|
> |MGM w/ DC|81.9|70.7|85.6|1573/355|92.0|49.8|
>
> **Q4. DC's performance on larger models**
>
> > It is speculated that the incremental performance gain might become even less pronounced when employing larger language models (LLMs)
>
> Thank you for highlighting this concern. We will address it from two aspects and hope our response alleviates the reviewers' concerns.
>
> - Due to limited resources, we previously used LoRA for fine-tuning, which affected the performance. Here, we provide a comparison of **the 34B model w/ and w/o the DC under the LoRA fine-tuning.** The results show that DC consistently enhances the model's performance in this scenario.
>
> ||TextVQA|MM-Bench|MMVet|
> |-|-|-|-|
> |LLaVA-1.5-34B-LoRA w/o DC|63.4|76.0|38.9|
> |LLaVA-1.5-34B-LoRA w/ DC|66.7|77.7|41.0|
>
> - We utilized more resources (32 A100 GPUs) to fully fine-tune 34B model, demonstrating that DC can achieve excellent results with larger models.
>
> ||TextVQA|MM-Bench|GQA|MMVet|LLaVA-Bench|
> |-|-|-|-|-|-|
> |DC-34B-AnyRes|75.2|81.2|66.6|59.2|97.7|
>
> **Q5.Bar charts for clearer comparisons.**
>
> Thanks. We provided bar charts in the PDF for clearer comparisons.
>
> **Q6. Combine multiple vision encoder**
>
> > DC can be further to combine multiple vision encoder, like DINOv2.
>
> We extended DC to multiple vision encoder, structures. Based on the results in [2] and our findings, we discovered that DINOv2 may not perform well. Therefore, in addition to the combination of CLIP and DINOv2, we also combined CLIP with ConvNeXT. The results show that the CLIP and ConvNeXT performs better.
>
> ||GQA|TextVQA|MMBench|
> |-|-|-|-|
> |CLIP+DINOv2|63.1|58.9|65.5|
> |CLIP+ConvNeXT|63.9|60.9|67.0|
>
> [2] Tong et al., Eyes Wide Shut? Exploring the Visual Shortcomings of Multimodal LLMs, CVPR 2024.

---

> > ### Comment · Reviewer_2pJ5 · 2024-08-12
> >
> > Thank you for the author's feedback.
> >
> > I recognize that there may have been some misunderstandings in my previous questions. To clarify, in addition to the original benchmarks, please include detailed comparisons for VQAv2, MME, POPE and etc  based on the MGM-13B framework.  Please further incorporate the original GQA, SQA^I, VQA^T, MMMU^v, and Math, as presented in Table 2 of the main paper.   Because I noticed a slight performance improvement on primary benchmarks: SQA^I (+0.3), GQA (+0.7) with the MGM-7B model, along with the POPE result (+0.5) on the MGM-13B.  Besides, I also noticed the performance gain of GQA (+0.6) on LLaVA-NeXT.
> >
> > Moreover, it would be beneficial to further provide detailed comparisons (approximately 10 benchmarks) with COMM, akin to extending the second table included in your response.
> >
> > I believe that a foundational module for MLLM should be robust enough to accommodate a variety of benchmarks while outperforming other similar methods.

---

> ### Author Response · Authors · 2024-08-13
> **Response to reviewer 2pj5**
>
> Thank you for the reviewer's comments. Due to the word limit, we only provided a portion of the benchmark results in our previous response. **We now offer a more comprehensive performance comparison.**
>
> **Q1.More comparisons with MGM**
>
> Thank you for your feedback. We agree that adding more benchmark comparisons will be helpful for this paper.
>
> We provide further comparisons between MGM-7B and MGM-13B here. According to the table below, DC improved performance across these benchmarks, **with an average increase of 1.1 on MGM-7B and 1.9 on MGM-13B.**
>
> |     | VQAv2 | MMBench | POPE | MME | LLaVA-Bench | MM-Vet | GQA | SQA | TextVQA | MMMU | MathVista | AVG |
> | --- | --- | --- | --- | --- | --- | --- | --- | --- | --- | --- | --- | --- |
> | MGM-7B | 80.4 | 69.3 | 85.7 | 1523/316 | 85.4 | 40.8 | 62.6 | 70.4 | 65.2 | 36.1 | 31.4 | 65.4 |
> | DC-7B  w/ MGM | 81.1 | 70.7 | 86.1 | 1530/347 | 88.7 | 42.2 | 63.3 | 70.7 | 66.0 | 36.8 | 32.5 | 66.5 (**+1.1%**) |
>
> |     | VQAv2 | MMBench | POPE | MME | LLaVA-Bench | MM-Vet | GQA | SQA | TextVQA | MMMU | MathVista | AVG |
> | --- | --- | --- | --- | --- | --- | --- | --- | --- | --- | --- | --- | --- |
> | MGM-13B | 80.8 | 68.5 | 85.1 | 1565/322 | 87.5 | 46.0 | 63.4 | 72.6 | 65.9 | 38.1 | 37.0 | 67.2 |
> | DC-13B  w/ MGM | 81.9 | 70.7 | 85.6 | 1573/355 | 92.0 | 49.8 | 64.2 | 74.9 | 66.7 | 39.3 | 38.1 | 69.1(**+1.9%**) |
>
> In current MLLMs, no matter which model we consider, there are inevitably instances where performance gains on certain benchmarks are minimal. Our experiments trained various models across different settings to validate the generalization of DC, including different vision encoders, LLMs, training data, modalities, and architectures. Given the diversity of settings, it's understandable that some results may appear less pronounced.
>
> We provided **official results** to show that even representative models (LLaVA, MGM) are unable to achieve significant improvements across all benchmarks. For example, in LLaVA-1.5, both the 7B and 13B models achieved the **same score of 85.9 on POPE**. In LLaVA-Next, the 7B to 13B models only improved by **+0.7** on POPE, **+0.7** on MathVista, and **+0.6** on MMMU. And **expanding LLaVA-1.5 13B to the high-resolution LLaVA-NeXT yielded only a 0.3 improvement on POPE.** Similarly, in MGM, when extended to higher resolution, MGM-13B showed improvements of **+0.1** on MMBench, **-0.8** on MMMU, and **+0** on Math.
>
> We want to convey that, like representative models in MLLM such as LLaVA and MGM, even **they cannot achieve consistent improvements across so many benchmarks**.
>
> We analyzed that, on one hand, the POPE is **relatively simple**, with results already being high, making further improvements more difficult. Additionally, POPE evaluates the model's ability to handle hallucinations, and since neither LLaVA nor DC is specifically designed to address this, the gains on POPE are both limited. On the other hand, benchmarks like SQA and GQA often **rely more on language capabilities than visual signals**. These benchmarks don't require strong visual perception to perform well on VQA tasks. For instance, **the test question in SQA below can be answered without any visual input.** Since DC is designed to enhance visual perception, its impact on SQA and GQA may be limited in these cases.
>
> > SQA: "question_id": "2828", "prompt": "<image>\nWhat is the capital of Iowa?\nA. Davenport\nB. Helena\nC. Lansing\nD. Des Moines\nAnswer with the option's letter from the given choices directly."
>
> DC has demonstrated noticeable performance gains across most benchmarks, achieved with almost **no additional overhead.** For benchmarks that **rely more on visual perception**, such as VQAv2, MM-Vet, and MMBench, DC improved MGM-13B by **1.1, 3.8, and 2.2**, respectively.
>
> **Q2.More comparisons with COMM**
>
> Thank you for your suggestion! Adding more benchmark comparisons will further strengthen this paper.
>
> We offer additional benchmark comparisons with COMM, showing that DC's average performance improved by **+1.4% over LLaVA** and by **+1.7% over COMM.**
>
> |     | TextVQA | MME | SQA | POPE | LLaVA-Bench | GQA | VQAv2 | MM-Vet | MM-Bench | MathVista | AVG |
> | --- | --- | --- | --- | --- | --- | --- | --- | --- | --- | --- | --- |
> | LLaVA (baseline) | 58.2 | 1511 | 66.8 | 85.9 | 65.4 | 61.9 | 78.5 | 31.1 | 64.3 | 24.9 | 61.3 |
> | LLaVA+COMM | 58.0 | 1493 | 67.4 | 86.0 | 60.5 | 62.9 | 79.2 | 30.1 | 65.7 | 25.8 | 61.0 **(-0.3%)** |
> | LLaVA+DC | 59.2 | 1511 | 69.5 | 86.6 | 66.1 | 63.8 | 79.5 | 32.7 | 66.8 | 26.9 | 62.7 **(+1.4%)** |
>
> We hope this result convinces the reviewer that DC is robust across various benchmarks and outperforms other similar methods.
>
> Thank you very much for your time and comments! Considering that the deadline for discussion is approaching, please let us know if there are any further questions that need clarification.

---

> > ### Comment · Reviewer_2pJ5 · 2024-08-13
> >
> > Thanks for the efforts for the further response.
> >
> > The results you've presented above thus far are convincing, and I think that they will enhance the robustness of the proposed method.
> >
> > The detailed and comprehensive comparisons above on the MGM-7B/13B, as well as its contrast with COMM, furnishes a validation of the approach's efficacy.  I think $\textbf{these results should be incorporated into the main body of the paper}$ to ensure that the readers are fully apprised of this work.
> >
> > I lean to the positive  and raise my score.

---

> > > ### Author Response · Authors · 2024-08-13
> > > **Response to reviewer 2pj5**
> > >
> > > We are very grateful for the reviewer's hard work. We believe that the reviewer's feedback has greatly refined our paper.
> > >
> > > We will revise the paper according to the reviewers' suggestions and **incorporate the aforementioned results into the main body of the paper.**

---

### Official Review · Reviewer_sj7M · 2024-07-14

**Soundness:** 4
**Presentation:** 4
**Contribution:** 3
**Rating:** 7
**Confidence:** 5

**Summary:**

This paper introduces the Dense Connector, a simple idea that aligns visual and language modalities by utilizing multi-layer visual features. The authors explore three instantiation methods and demonstrates the effectiveness of the Dense Connector across various settings, including different backbones, modalities, datasets and architectures.

**Strengths:**

1.	This paper focuses on further exploring the utilization of visual signals in MLLMs. Compared to the attention given to language models, this perspective indeed lacks sufficient attention. It provides insights into utilizing MLPs for integrating multi-layer visual features at the channel level in MLLMs, reducing visual tokens and enhancing computational efficiency.
2.	Dense Connector demonstrates effectiveness across a range of model architectures, including LLaVA and Mini-Gemini, as well as LLMs scaling from 2.7B to 70B parameters.
3.	This paper is well-written and includes thorough experimental validation of the Dense Connector.
4.	I also tried running the code provided by the author, and it is indeed simple and effective. Personally, I feel dense connector has the potential to become a fundamental module in future MLLMs.

**Weaknesses:**

1.	High-resolution approaches are widely utilized in vision-language models. This paper explores the application of the Dense Connector with dual visual encoders, specifically Mini-Gemini, in high-resolution settings. However, it lacks an exploration of dynamic high-resolution scenarios, such as those implemented by LLaVA-NEXT. I am curious about whether the Dense Connector could further enhance MLLM performance in this scenario.
2.	My another concern is that although the Dense Connector performs well on the Vicuna 7B and 13B models, its effectiveness seems to diminish when scaling to larger sizes. I suspect this might be because training LoRA is not as effective as updating the LLM. If possible, the authors should provide results for updating the LLM to give readers a better reference.

**Questions:**

Beyond the section on weaknesses, I have the following additional questions:
1.	The decision to fine-tune ViT during MLLM training is still debated. Research findings are mixed; some [1] indicate that fine-tuning ViT may impair performance, while others [2] show it can lead to performance improvements. I wonder whether the Dense Connector, which utilizes multi-layer visual features, could benefit from fine-tuning ViT.
2.	The authors could specify the additional training and testing time required by the Dense Connector.
[1] Karamcheti, Siddharth, et al. "Prismatic vlms: Investigating the design space of visually-conditioned language models." arXiv preprint arXiv:2402.07865 (2024).
[2] Chen, Lin, et al. "Sharegpt4v: Improving large multi-modal models with better captions." arXiv preprint arXiv:2311.12793 (2023).

**Limitations:**

The authors have addressed the limitations.

---

> ### Author Rebuttal · Authors · 2024-08-06
>
> **Q1. Combining Dense Connector with LLaVA-NEXT's AnyRes technology**
>
> Thanks! This is a very worthwhile question to explore. We extended the Dense Connector (DC) to dynamic resolution scenarios. Using dynamic resolution, DC remains effective. Compared to the baseline, DC improved performance on TextVQA and MMMU by **1.1\%** and **2.2%**, respectively. Additionally, **using only LLaVA-1.5 data, DC outperformed LLaVA-NeXT on several benchmarks**.
>
> | Model | Data | Vision Encoder | LLM | TextVQA | SQA | GQA | LLaVA-Bench | MM-Bench | MM-Vet | MMMU |
> | --- | --- | --- | --- | --- | --- | --- | --- | --- | --- | --- |
> | LLaVA-NeXT (Baseline) | LLaVA-1.5 | CLIP-L/336px | Vicuna-7B | 64.5 | 69.5 | 64.0 | 68.2 | 66.5 | 33.1 | 35.4 |
> | DC  | LLaVA-1.5 | CLIP-L/336px | Vicuna-7B | 65.6 | 70.5 | 64.6 | 69.0 | 67.4 | 33.7 | 37.6 |
> | DC  | MGM | SigLIP-SO | Vicuna-7B | 70.0 | 72.0 | 63.9 | 88.8 | 69.2 | 44.4 | 35.8 |
> | DC  | MGM | SigLIP-SO | Vicuna-13B | 70.9 | 75.2 | 64.3 | 93.2 | 72.3 | 47.0 | 35.8 |
>
> **Q2. Explanation regarding the performance of larger language models**
>
> Thank you for raising this concern. We will address it from two aspects and hope our response alleviates the reviewers' worries.
>
> - In previous experiments, constrained by limited computational resources, we **used LoRA to reduce memory consumption** for fine-tuning the 34B model. Here, we present a comparison of the 34B model w/ and w/o the DC under LoRA fine-tuning. The results indicate that the inclusion of DC consistently enhances the model's performance in this context.
>
>
> |     | TextVQA | MM-Bench | MM-Vet |
> | --- | --- | --- | --- |
> | LLaVA-1.5-34B-LoRA w/o DC | 63.4 | 76.0 | 38.9 |
> | LLaVA-1.5-34B-LoRA w/ DC | 66.7 | 77.7 | 41.0 |
>
> - Now, with more computational resources, we **fully fine-tuned the 34B model**, addressing concerns about DC's performance on larger language models. The results are shown in the table below.
>
>
> |     | Data | Vision Encoder | Res. | LLM | TextVQA | SQA | GQA | LLaVA-Bench | MM-Bench | MM-Vet | MMMU |
> | --- | --- | --- | --- | --- | --- | --- | --- | --- | --- | --- | --- |
> | DC  | MGM | SigLIP-SO | AnyRes | Yi-34B | 75.2 | 78.0 | 66.6 | 97.7 | 81.2 | 59.2 | 47.2 |
>
> **Q3.Will fine-tuning ViT improve performance?**
>
> Thanks! That's a great question. We fine-tuned ViT using the settings from LLaVA-NeXT, specifically a learning rate of 2e-6 in the second stage. The results indicate that **fine-tuning ViT can further improve the performance of DC**.
>
> | Model | Data | Vision Encoder | Fine-tune ViT | Res. | TextVQA | VQAv2 | SQA | GQA | LLaVA-Bench | MM-Bench | MM-Vet |
> | --- | --- | --- | --- | --- | --- | --- | --- | --- | --- | --- | --- |
> | DC-7B | LLaVA-1.5 | CLIP-L | No  | 336 | 59.2 | 79.5 | 69.5 | 63.8 | 66.1 | 66.8 | 32.7 |
> | DC-7B | LLaVA-1.5 | CLIP-L | Yes | 336 | 60.2 | 80.5 | 68.4 | 63.7 | 66.2 | 68.6 | 34.4 |
>
> | Model | Data | Vision Encoder | Fine-tune ViT | Res. | TextVQA | VQAv2 | SQA | GQA | LLaVA-Bench | MM-Bench | MM-Vet |
> | --- | --- | --- | --- | --- | --- | --- | --- | --- | --- | --- | --- |
> | DC-7B | LLaVA-1.5 | SigLIP-SO | No  | AnyRes | 66.5 | 81.4 | 69.3 | 64.8 | 70.7 | 67.2 | 34.8 |
> | DC-7B | LLaVA-1.5 | SigLIP-SO | Yes | AnyRes | 67.6 | 82.5 | 69.2 | 64.8 | 71.1 | 68.9 | 35.0 |
>
> **Q4. About training time and inference time**
>
> For the Sparse Token Integration method, the addition of visual tokens results in a 20% increase in training and inference time. However, for Sparse Channel Integration and Dense Channel Integration, the training and inference times **remain unchanged**. Specifically, using 8 A100 GPUs (40G) to train the Vicuna-7B model, DC takes approximately **4 hours** in the first stage and **12 hours** in the second stage, **similar to LLaVA-1.5**.

---

### Author Rebuttal · Authors · 2024-08-06

We sincerely thank all the reviewers for their time and insightful comments. We are glad the reviewers find that Dense Connector (DC) is a novel plug-and-play module, which is the first in the MLLM field to explore the fusion of multi-layer visual features from both token and channel dimensions. We introduced three fusion methods: STI, SCI, and DCI, all of which achieved significant performance improvements. We are encouraged that the reviewers acknowledge our extensive experiments demonstrating its effectiveness across different encoders, resolutions, model frameworks, LLMs, training data, and modalities (sj7M, 2pJ5, qnn9). We are also delighted that the reviewers tried using our module and found it effective (sj7M).

**[Paper improvements made in response to feedback]**

In response to the reviewers' valuable suggestions, we have made the following improvements to the paper:
- We have extended DC to more architectures, including dynamic resolution methods **LLaVA-NeXT** and cross-attention **visual resamplers**, demonstrating that DC can be widely applied in MLLMs (sj7M, qnn9).
- Our experiments show that **fine-tuning ViT can benefit DC**, which utilizes multi-layer features (sj7M).
- We have validated that our method is **superior to previous methods** that utilize multi-layer features (2pJ5).
- We extended DC to the high-resolution method MGM-13B, where **DC significantly improved MGM-13B's performance** (2pJ5).
- By **fully fine-tuning the 34B model**, we hope this result alleviates reviewers' concerns about DC's performance on larger models (2pJ5, sj7M).
- We have also validated that DC can be applied to **multiple vision encoder architectures** (2pJ5).
- We have included **charts** in the PDF, which provides a clearer visual comparison (2pJ5).

We did our best to address the questions within the given time. We appreciate the reviewers' comments and believe the revisions have strengthened the paper. We thank all the reviewers for their help. Please find individual responses to your questions below.

---

### Decision · Program_Chairs · 2024-09-25

**Decision:**

Accept (poster)

**Comment:**

This work intoduced dense connector for multimodal LLMs to enhance to the visual percetion capability, akin to densenet or FPN in vision models. The method is simple yet effective across the board, spanning from different benchmarks with different resolutions and variou LLM sizes. The paper is also well-written and method is clearly motivated. During the discussion, the authors and reviewers were engaged in intensive discussions. The authors provided a detailed and constructive rebuttal, addressing all reviewer concerns effectively. As a result, it recevied all positive final ratings (5, 6, 7).

Overall, this work presents a technically sound contribution to the field of MLLMs. According to the submission and discussions, the proposed Dense Connector is a versatile module that offers consistent performance improvements with minimal additional computation. While some concerns were raised about the incremental nature of the gains and the generalizability of the approach, the authors’ thorough rebuttal addressed these issues convincingly. As a result, the ACs recommend an acceptance, and suggest the authors integrate the additional experimental results and analysis during rebuttal to the final version, and release the implementation code to public for benefit to the community.